

# Five new pseudocryptic land planarian species of *Cratera* (Platyhelminthes: Tricladida) unveiled through integrative taxonomy

Ana Paula Goulart Araujo[1,2], Fernando Carbayo[2,3], Marta Riutort[4] and Marta Álvarez-Presas[4]

[1] Museu de Zoologia, Universidade de São Paulo, São Paulo, Brazil
[2] Laboratório de Ecologia e Evolução, Escola de Artes, Ciências e Humanidades, Universidade de São Paulo, São Paulo, Brazil
[3] Departamento de Zoologia, Instituto de Biociências, Universidade de São Paulo, São Paulo, Brazil
[4] Departament de Genètica, Microbiologia i Estadística, Facultat de Biologia and Institut de Recerca de la Biodiversitat (IRBio), Universitat de Barcelona (UB), Barcelona, Spain

Corresponding author
Marta Álvarez-Presas,
onaalvarez@ub.edu

## ABSTRACT

**Background.** *Cratera* is a genus of land planarians endemic to the Brazilian Atlantic forest. The species of this genus are distinguished from each other by a series of external and internal characters, nonetheless they represent a challenging taxonomic issue due to the extreme alikeness of the species analysed in the present work. To resolve these difficulties, we have performed morphological analyses and used three nuclear markers (ribosomal 18S and 28S, Elongation Factor, a new anonymous marker named Tnuc813) and two mitochondrial fragments (Cytochrome c oxidase subunit I gene, and a fragment encompasing NADH deshydrogenase subunit 4 gene, trnF and the beginning of the Cytochrome c oxidase I gene) in an integrative taxonomic study.

**Methods.** To unveil cryptic species, we applied a molecular species delimitation approach based on molecular discovery methods, followed by a validation method. The putative species so delimited were then validated on the basis of diagnostic morphological features.

**Results.** We discovered and described four new species, namely *Cratera assu, C. tui, C. boja,* and *C. imbiri.* A fifth new species, *C. paraitinga* was not highly supported by molecular evidence, but was described because its morphological attributes are unique. Our study documents for the genus *Cratera* the presence of a number of highly similar species, a situation that is present also in other genera of land planarians. The high number of poorly differentiated and presumably recent speciation events might be explained by the recent geological history of the area.

## INTRODUCTION

Land planarians (Platyhelminthes: Tricladida: Geoplanidae) are mostly soil inhabitants of forested areas. There are over 900 known species (*Sluys, 2016*), 332 of them belonging to Geoplaninae (http://planarias.each.usp.br; accessed in 18. March 2020), an exclusively

Neotropical subfamily. Anatomy and histology of the copulatory apparatus are central for the identification and systematics of these organisms (e.g., *Froehlich, 1955*; *Negrete & Brusa, 2016*). Nonetheless, when in several studies traditional, morphology-based taxonomic approaches were complemented with molecular methodologies, some nominal species were found to be polyphyletic (*Sluys et al., 2016*; *Carbayo et al., 2018*; *Almeida, Marques & Carbayo, 2019*). Detailed reanalyses of the morphological evidence in those cases revealed that morphological variation assumed to represent within-species polymorphisms, actually signaled the existence of distinct species. From another perspective, such reinterpretation of intra-specific morphological variation revealed the presence of pseudocryptic species (see references above; *Sáez & Lozano, 2005*).

The systematics of the Geoplaninae above the species level has benefited also from the molecular approach. Molecular phylogenetic analyses of this group revealed a number of polyphyletic genera. One of these genera, *Geoplana* Stimpson, 1857 subsequently being split into several genera (*Carbayo et al., 2013*). The genus *Cratera Carbayo et al., 2013* emerged from *Geoplana* as a monophyletic group with nine species to which were gradually added another 11 species with similar features (*Carbayo & Almeida, 2015*; *Negrete & Brusa, 2016*; *Rossi et al., 2014*; *Rossi et al., 2016*; *Rossi & Leal-Zanchet, 2017*; *Lago-Barcia & Carbayo, 2018*; *Boll, Amaral & Leal-Zanchet, 2019*). The most conspicuous diagnostic feature of *Cratera* is an ejaculatory duct with its distal section being widened (*Marcus, 1951*; *Lago-Barcia & Carbayo, 2018*). However, this trait is not present in all members of the genus, probably as a result of secondary loss (*Lago-Barcia & Carbayo, 2018*).

In the course of extensive land planarian samplings across the Atlantic forest we have found many individuals that can be attributed to the genus *Cratera*, most of them presenting very similar or even identical features in their external aspect or their internal anatomy. Given the presence of cryptic or pseudocryptic species in other land planarian genera, we applied an integrative taxonomic analysis to unveil eventual cryptic species. We adopted the General Lineage Species Concept, which defines species as independently evolving metapopulation lineages (*De Queiroz, 1998*). In order to implement this concept, we used an integrative approach to species delimitations. First, we applied molecular species delimitation methods to delineate Primary Species Hypotheses (PSH) based on discovery methods, and thereafter used a validation method to formulate Secondary Species Hypotheses (SSH; *Puillandre et al., 2012a*). Hereafter, we tested whether the putative species exhibited morphological or anatomical features supporting their validity. In this manner, we unveiled four species for which molecular and morphological data agreed with each other. Molecular data of a putative fifth species did not fully support its distinctness, but morphological data did clearly point to its separate specific status and therefore we gave priority to the latter source of evidence.

## MATERIALS & METHODS

### Specimens sampling and morphological study

Intensive samplings were performed in four protected forest areas in the States of Rio de Janeiro and São Paulo (Fig. 1). A number of people ranging between 5–8 performed

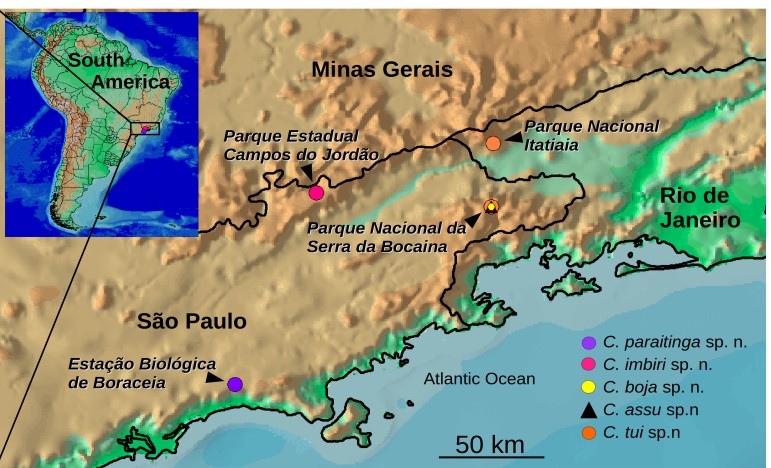

**Figure 1** **Map showing the sampling sites.** Records of the five species of *Cratera* described in this paper. Map generated with DIVA-GIS 7.5 (http://swww.diva-gis.org/Data).

each of two to 8-day field campaigns until reaching 200 h sampling in each area (with the exception of Campos do Jordão, with 75 h). Field experiments were approved by COTEC - Instituto Florestal do Estado de São Paulo (Proc. SMA 12.640/2011), Museu de Zoologia (EBBAut.020/2013) and Instituto Chico Mendes de Conservação da Biodiversidade (Proc. 32779-1; 11748-4). Animals were collected from the soil litter during the day and at night. The worms were photographed and, subsequently, killed in boiling water, after which a small tissue sample was taken and preserved in 100% ethanol for DNA extraction. Vouchers of frozen tissues are kept in FC's laboratory. The remaining part of the body was fixed in 10% formalin and, subsequently, transferred to 80% ethanol. Parts of the body were embedded in paraffin Histosec®, sectioned at intervals of 2–7 μm, mounted on glass slides, and subsequently stained with Mallory method as modified by *Cason (1950)*. Slides were examined with a compound microscope. Reconstruction drawings were done with a camera lucida attached to the microscope. Photomicrographs were taken with the help of a digital camera attached to the microscope. Enhancement of the contrast of the microphotographs and a whitish background of the photomicrographs were done with GIMP (GNU Image Manipulation Program 2.8.16; The GIMP team http://www.gimp.org, 1995–2016). Descriptions of the body color of live or preserved specimens follow the online RAL palette colors (©RAL gemeinnützige GmbH, available at https://www.ral-farben.de/uebersicht-ral-classic-farben.html?&L=1). Drawings and photomicrographs of sagittal and horizontal views are orientated with anterior extremity to the left. The width of the creeping sole was measured on transverse sections of the pre-pharyngeal region. Type material was deposited in the Museu de Zoologia da Universidade de São Paulo (MZUSP).

The electronic version of this article in Portable Document Format (PDF) will represent a published work according to the International Commission on Zoological Nomenclature (ICZN), and hence the new names contained in the electronic version are effectively

published under that Code from the electronic edition alone. This published work and the nomenclatural acts it contains have been registered in ZooBank, the online registration system for the ICZN. The ZooBank LSIDs (Life Science Identifiers) can be resolved and the associated information viewed through any standard web browser by appending the LSID to the prefix http://zoobank.org/. The LSID for this publication is: urn:lsid:zoobank.org:pub:F6B30CB7-6114-434F-9B2A-A2F4CE625A20. The online version of this work is archived and available from the following digital repositories: PeerJ, PubMed Central and CLOCKSS.

## Molecular data acquisition

Extractions of genomic DNA were performed using the Wizard® Genomic DNA Purification kit (Promega, Madison, WI, USA) following *Álvarez Presas et al. (2011)*. Two mitochondrial and four nuclear markers were selected. The mitochondrial markers are a cytochrome oxidase I gene fragment (hereafter referred to as COI), and a mitochondrial fragment which includes the end of the NADH dehydrogenase subunit 4 gene (nad4), the entire sequence of trnF and the beginning of the COI (cox1) gene. This latter marker, hereafter referred to as Nd4toCox1, is tested for the first time in this work. The four nuclear genes correspond to the 18S rDNA type II (18S), a fragment of the 28S rDNA (28S), a partial coding region of the elongation factor 1-alpha gene (hereafter referred to as EF), and an anonymous nuclear marker (hereafter referred to as Tnuc813) developed from NGS data (as detailed in *Leria et al., 2020*) and tested here for the first time. Primers used to amplify and sequence the genes are mentioned in Table S1. For some individuals (indicated in Table 1), it was not possible to obtain a long COI sequence (∼900 bp) of good quality (fragment amplified by the BarS/COIR primers). To overcome this situation, a shorter fragment (COIF/COIR primers), of ∼300 bp, was amplified. The polymerase chain reaction (PCR) amplification (25 μL) was performed on a Techne® TC-5000TM (Bibby Scientific Ltd, Staffordshire, UK) and on an Eppendorf Mastercycler® (Eppendorf, Hamburg, Germany) personal thermocyclers using initial denaturation step of 5 min at 92–95 °C, followed by 30–35 cycles of 30 to 50 s denaturation at 94–95 °C, 30 to 45 s annealing at 44–54 °C and 50 s –1 min extension at 72 °C, with a final extension step of 3–4 min at 72 °C. The PCR results were verified using electrophoresis of the amplification products on 1% agarose gels stained with GelRed (Biotium, Hayward, CA, USA), and visualized under UV transillumination. Amplification products were purified with a vacuum manifold (Multiscreen®HTS Vacuum Manifold; Millipore Corporation, Billerica, MA, USA). Purification products were sent to Macrogen (Amsterdam, Europe), where both strands were sequenced by Sanger sequencing. Chromatograms were revised and contigs constructed in Geneious v 8.1.7. software (Biomatters; available from http://www.geneious.com/).

For all the coding genes (COI, Nd4toCox1, EF and Tnuc813), sequences were aligned based on the amino acid sequences by using Clustal W (included in the BioEdit software 7.0.9.0 (*Hall, 1999*)). The genetic code 9 (Echinoderm and flatworms' mitochondrial) was used for translating the mitochondrial genes. Ribosomal RNA gene sequences were aligned using the online version of the software Mafft v7 (*Katoh, Rozewicki & Yamada, 2017*) applying the G-INS-i iterative refinement method. Misaligned or ambiguously aligned

Araujo et al. (2020), *PeerJ*, DOI 10.7717/peerj.9726

**Table 1** **List of *Cratera* samples used in this study with sampling locality, voucher code and GenBank accession numbers.**

| Species | Sampling locality | Museum code | GenBank accession number | | | | | |
|---------|-------------------|-------------|------|---------|-----|-----|-----|---------|
| | | | COI | Nd4Cox1 | 18S | 28S | EF | Tnuc813 |
| *Cratera arucuia* | P.N. Intervales / SP | MZUSP [a] PL 1048 | KC608281[c] | MT468629[*] | KC608513 | KC608396 | KC614508 | MT468607[*] |
| *Cratera crioula* | P.E. Serra da Cantareira / SP | MZUSP PL 471 | MT437776[*] | – | – | – | – | – |
| | P.N. Bocaina / SP | MZUSP PL 0459 | KU564215 | – | – | – | – | – |
| | P.E. Serra da Cantareira / SP | MZUSP PL 1078 | KC608323 | – | KC608557 | KC608440 | KC614543 | MT468615[*] |
| | | MZUSP PL 1079 | KC608324 | – | KC608558 | KC608441 | KC614544 | MT468616[*] |
| *Cratera cuarassu* | P.E. do Desengano / RJ | MZUSP PL 348 | MT437766[*] | MT468626[*] | KC608510 | KC608393 | KC614505 | MT468603[*] |
| | | MZUSP PL 349 | MT437767[*] | MT468627[*] | MT441688[*] | MT441711[*] | MT468580[*] | MT468604[*] |
| | | MZUSP PL 350 | MT437768[*] | – | MT441689[*] | MT441712[*] | MT468581[*] | MT468605[*] |
| | | MZUSP PL 351 | MT437769[*] | MT468628[*] | MT441690[*] | MT441713[*] | MT468582[*] | MT468606[*] |
| | | MZUSP PL 805 | MT437777[*] | – | – | – | – | – |
| | | MZUSP PL 806 | MT437778[*] | – | – | – | – | – |
| | P.E. do Desengano / RJ | MZUSP PL 807 | MT437779[*] | – | – | – | – | – |
| | | MZUSP PL 808 | MT437780[*] | – | – | – | – | – |
| *Cratera imbiri* sp. nov. | P.E. Campos de Jordão / SP | MZUSP PL 2155 | MT437782[*] | – | MT441697[*] | MT441720[*] | MT468589[*] | MT468618[*] |
| *Cratera ochra* | | MZU PL.00192 | KT250624[c] | – | – | – | – | – |
| | | MZU PL.00191 | KT250623 | – | – | – | – | – |
| | | MZU PL. 1564 | KT250622 | – | – | – | – | – |
| *Cratera paraitinga* sp. nov. | Estação Bio. Boracéia /SP | MZUSP PL 2156 | MT437783[*c] | MT468634[*] | MT441698[*] | MT441721[*] | MT468590[*] | MT468619[*] |
| | | MZUSP PL 2157 | MT437784[*c] | MT468635[*] | MT441699[*] | MT441722[*] | MT468591[*] | MT468620[*] |
| *Cratera picuia* | P.N. Saint Hilaire / PR | MZUSP PL 1008 | KC608261 | – | KC608493 | KC608376 | KC614491 | MT468598[*] |
| *Cratera boja* sp. nov. | P.N. Bocaina / SP | MZUSP PL 458 | MT437773[*] | – | MT441693[*] | MT441716[*] | MT468585[*] | – |
| | P.N. Bocaina / SP | MZUSP PL 459 | MT437774[*] | – | MT441694[*] | MT441717[*] | MT468586[*] | – |
| *Cratera assu* sp. nov. | P.N. Bocaina / SP | MZUSP PL 2146 | MT437763[*] | – | MT441685[*] | MT441708[*] | MT468577[*] | MT468599[*] |
| | | MZUSP PL 1050 | KC608284 | – | KC608516 | KC608399 | KC614510 | MT468609[*] |
| | | MZUSP PL 1052 | KC608287 | – | KC608519 | KC608402 | KC614513 | MT468611[*] |
| | | MZUSP PL 2150 | MT437772[*] | MT468631[*] | MT441692[*] | MT441715[*] | MT468584[*] | MT468612[*] |
| | | MZUSP PL 2151 | MT437775[*] | MT468632[*] | MT441695[*] | MT441718[*] | MT468587[*] | MT468613[*] |

Araujo et al. (2020), *PeerJ*, DOI 10.7717/peerj.9726

**Table 1** (*continued*)

| | | | GenBank accession number | | | | | |
|---|---|---|---|---|---|---|---|---|
| **Species** | **Sampling locality** | **Museum code** | **COI** | **Nd4Cox1** | **18S** | **28S** | **EF** | **Tnuc813** |
| *Cratera tui* sp. nov. | P.N. Bocaina / SP | MZUSP PL 1014 | KC608268[c] | MT468625[*] | KC608500 | KC608383 | KC614497 | MT468600[*] |
| | | MZUSP PL 2147 | MT437764[*c] | – | MT441686[*] | MT441709[*] | MT468578[*] | MT468601[*] |
| | | MZUSP PL 2148 | MT437765[*] | – | MT441687[*] | MT441710[*] | MT468579[*] | MT468602[*] |
| *Cratera tui* sp. nov. | P.N. Bocaina / SP | MZUSP PL 2149 | MT437770[*c] | – | MT441691[*] | MT441714[*] | MT468583[*] | MT468608[*] |
| | | MZUSP PL 1051 | MT437771[*c] | MT468630[*] | KC608517 | KC608400 | KC614511 | MT468610[*] |
| | P.N. Itatiaia / RJ | MZUSP PL 2154 | MT437781[*] | MT468633[*] | MT441696[*] | MT441719[*] | MT468588[*] | MT468617[*] |
| *Cratera pseudovaginuloides* | P.N. Órgaõs / RJ | MZUSP PL 670 | KC608251 | MT468622[*] | KC608483 | KC608366 | KC614482 | MT468593[*] |
| | | MZUSP PL 671 | KC608252 | – | KC608484 | KC608367 | KC614483 | MT468594[*] |
| *Cratera tamoia* | P.N. Órgaõs / RJ | MZUSP PL 665 | KC608246 | MT468621[*] | KC608478 | KC608361 | KC614478 | MT468592[*] |
| | | MZUSP PL 672 | KC608254 | MT468623[*] | KC608486 | KC608369 | KC614484 | MT468595[*] |
| **Outgroup** | | | | | | | | |
| *Obama anthropophila* | P.N. Itajaí / SC | MZUSP PL 1007 | KC608256 | MT468624[*] | KC608488 | KC608371 | KC614486 | MT468596[*] |
| *Obama ladislavii* | | MZUSP PL 681 | KC608258 | – | KC608490 | KC608373 | KC614488 | MT468597[*] |
| *Obama josefi* | FLONA[b] | MZUSP PL 1075 | KC608318 | – | KC608552 | KC608435 | KC614538 | MT468614[*] |

**Notes.**
[a] Vouchers are deposited in the Museu de Zoologia da Universidade de São Paulo (MZUSP).
[b] Floresta Nacional de São Francisco de Paula.
[c] Short fragment.
*This study.

regions were removed using Gblocks v0.91b program (*Talavera & Castresana, 2007*), allowing 50 as a maximum number of contiguous non-conserved positions and setting the minimum length of a block to 4, and allowing half gap positions allowed. Three different datasets were used for several analyses: (1) *COI* dataset including COI sequences used for the ABGD and mPTP molecular species delimitation approaches; (2) *BPP* datasets 18S, 28S, COI, Nd4toCox1, Tnuc813, and EF independent alignments (completing some sequences with missing data (Ns); see Table 1) used for the BPP molecular species delimitation analysis; and (3) *concatenated* dataset, including the information of the six genes (18S, 28S, EF, Tnuc813, COI and Nd4toCox1), which was used to infer a general phylogeny.

For the individual gene alignments, the DNA sequence evolution model that best fits the data was estimated by using jModelTest v2.1.4 (*Darriba et al., 2012*), applying the Akaike information criterion (AIC). For the concatenated dataset PartitionFinder2 version 2.1.1 (*Lanfear et al., 2017*) was run on the CIPRES Science Gateway (*Miller, Pfeiffer & Schwartz, 2010*) to identify an appropriate partition scheme and their corresponding DNA evolutionary models. The data were divided by gene, with unlinked branch lengths, the 'raxml' models for selection and the AICc model selection criteria with the 'greedy' search algorithm. The phylogenetic trees for the concatenated dataset were inferred using the Bayesian Inference (BI) method using MrBayes software v3.2.6. (*Ronquist et al., 2012*) implemented in CIPRES and using BEAGLE (*Ayres et al., 2012*), setting the evolutionary model and appropriate partitions according to the PartitionFinder results with the unlinked parameters. Two runs of four chains were applied producing 5 million generations and, for each of them, 5,000 trees were stored. It was checked that the probability values (logarithm) of the cold chain reached the stationarity state and the convergence of the two runs, verifying that the average standard deviation of the split frequencies was lower than 0.01. A default burn-in of 25% was used and a consensus tree was obtained from the remaining trees. The maximum likelihood (ML) method was used to infer phylogenies with the software IQtree v1.6.10 (*Nguyen et al., 2015*). The IQtree searches were carried out using the default configuration in CIPRES, with a starting random tree (option -t RANDOM), and assessing branch support using 1,000 ultrafast bootstrap approximation replicates (*Minh, Nguyen & Von Haeseler, 2013*). The best fit models for each partition were selected by PartitionFinder and each partition was allowed to have its own set of branch lengths (option -sp).

## Molecular species delimitation

For the molecular species delimitation analyses, two discovery methods (ABGD and mPTP) and one validation method (BPP) were applied. Using the COI dataset, the Automatic Barcode Gap Discovery (ABGD) method (*Puillandre et al., 2012b*) was applied through the website http://wwwabi.snv.jussieu.fr/public/abgd/abgdweb.html. The default values of Pmin = 0.001 and Pmax = 0.10, steps = 10 and number of intervals = 20 were used, while also the relative gap width value (X) = 1.0 and correcting the distance matrix under the K80 Kimura model with a MinSlope = 1.5.

The multi-rate Poisson Tree Process (mPTP), which is another single locus analysis, was also used. This model incorporates different levels of intraspecific genetic diversity derived from differences in the evolutionary history or in the sampling of each species,

accommodating different coalescence rates within the lineages (*Kapli et al., 2017*). mPTP analysis was performed in a ML tree reconstructed by IQtree in CIPRES with the *COI* dataset (Suppl Fig. 1). For this analysis the command line version of the mPTP v 0.2.4. software was used without considering the outgroup. Four independent runs of 5,000,000 Monte Carlo Markov chains (MCMC) were carried out, sampling every 10,000 generations. The use of these discovery methods leads to the Primary Species Hypothesis (PSH), used as starting point for the validation step.

For the validation step, a Bayesian multilocus method of delimiting species (*Yang & Rannala, 2010*; *Yang & Rannala, 2014*) implemented in the BPP v3.3 software (*Yang, 2015*) was applied. Different hypotheses of species delimitation and estimation of the posterior probability (PP) of each model were tested using reversible jump MCMC (rjMCMC). The previous species assignment resulting from the ABGD discovery analysis was used as a starting hypothesis for the BPP analysis, because it was the analysis that gave the largest number of PSHs. Some species were excluded from this validation analysis because only one individual was available (*Cratera arucuia Lago-Barcia & Carbayo, 2018* and *Cratera picuia Lago-Barcia & Carbayo, 2018*) or because it only had the COI gene sequenced (*Cratera ochra Rossi et al., 2016*). As these three species were not the target species for our study, their removal from the analysis was not relevant. A guide tree generated by 100 million generations (stored every 5000) in *BEAST2 v2.5.2 (*Bouckaert et al., 2014*) was built in CIPRES with the six single gene datasets (*BPP* datasets), applying the evolutionary model for each gene resulting from the previous jmodeltest analysis (18S=GTR+I; 28S=HKY+I+G; Cox1=GTR+G; EF=GTR+I+G; Nd4toCox1=HKY+I+G and Tnuc=GTR+I).

The molecular clock was set as log-normal-relaxed for all markers (unlinked) and the speciation model to Birth and Death. In the BPP analysis, both the size of the ancestral population (theta, $\theta$) and the time of origin for each species (tau, $\tau$) were parameterized with four different models (named M1-M4): M1 for large ancestral population size and deep divergence, G (1, 10) for $\theta$ and $\tau$; M2 for small ancestral population size and shallow divergence, G (2, 1000) for $\theta$ and $\tau$; M3 for large ancestral population size and shallow divergence, G (1 10) for $\theta$ and G (2 1000) for $\tau$; and M4 for small ancestral population size and deep divergence, G (2 1000) for $\theta$ and G (1 10) for $\tau$. The rjMCMC analysis was run under the Algorithm 0 in 100,000 generations (with a sampling interval of 2) excluding 10% as burn-in. In order to test the robustness of the results, these executions were replicated using different starting seeds. These analyses were done without outgroup. Results of BPP lead to Secondary Species Hypotheses (SSH).

## RESULTS

### Molecular datasets

The *COI* dataset consists of 40 sequences with a length of 822 bp, while the concatenated dataset (28 *Cratera* sequences plus 3 outgroups from the genus *Obama*) has a length of 5,671 bp. The individual gene datasets for the *BEAST analysis are constituted by 26 sequences with a length of 1,349 bp (18S), 1,544 bp (28S), 825 bp (COI), 612 bp (EF), 730 bp (Nd4toCox1) and 611 bp (Tnuc813).

## Phylogenetic analysis

The partitions obtained with PartitionFinder and applied to the phylogenetic analysis of the *concatenated* dataset are Cox1_codon2, EF_codon2, 18S, Tnuc813-1_codon3, Tnuc813-3_codon2, Tnuc813-3_codon3, Tnuc813-5_codon1, Tnuc813-5_codon2, Tnuc813-5_codon3 = K81UF + G, 28S = TIM + G, Cox1_codon1, EF_codon1 = TRN + G, Cox1_codon3 = GTR + G, EF_codon3, Tnuc813-1_codon2, Tnuc813-3_codon1 = TVM + G, Nd4toCox1-1_codon1, Nd4toCox1-1_codon2, Nd4toCox1-3_codon1, Nd4toCox1-3_codon3, Tnuc813- 1_codon1 = GTR + G, Tnuc813-2, Tnuc813-4 = GTR + G, Nd4toCox1-1_codon3, Nd4toCox1-2, Nd4toCox1-3_codon2 = HKY + G. For the concatenated analyses, the topologies obtained are the same in both methods (ML and BI), as shown in Fig. 2. There is only a small difference in the relationships between the specimens within two of the new species described in the present paper (*Cratera tui* sp. nov. and *C. assu* sp. nov.), but without statistical support for any of the two methods. All known species are monophyletic. Three of the new species described here (*C. tui* sp. nov., *C. imbiri* sp. nov. and *C. paraitinga* sp. nov.) form a monophyletic group, which in turn is sister to the species *C. cuarassu* (*Carbayo & Almeida, 2015*). *Cratera pseudovaginuloides* (*Riester, 1938*) is a sister lineage of *C. boja* sp. nov., both constituting the sister group of a large clade formed by the species *C. assu* sp. nov. and *C. crioula* (*Froehlich, 1955*) on the one hand and the group constituted by *C. picuia*, *C. arucuia* and *C. tamoia* (*Froehlich, 1955*). The vast majority of the relationships present high support values at the nodes for both methodologies (PP and bootstrap values) (Fig. 2).

## Molecular species delimitation

Concatenated tree (Fig. 2) provides a summary of the species delimitation results. The ABGD method delimits within the genus *Cratera* 13 Molecular Operational Taxonomic Units (MOTUs) with a *p*-value of P 0.001 - 0.0046. The mPTP method delimits 11 candidate species for the ingroup with average support of 0.90. In both delimitations, mPTP and ABGD, *C. ochra* is delimited as a single candidate species. However, it is not present in the concatenated tree because there are only COI sequences in GenBank for this species (Table 1 and Fig. S1). For the same reason, the species *C. ochra* was not included in the BPP analyses.

ABGD predicts a higher number of candidate species than mPTP. Therefore, the ABGD set of candidate species is adopted as a reference to designate the PSHs in order to test as many scenarios as possible in the validation with the BPP method. The assignment is as follows. All known species are delimited by both ABGD and mPTP methods as candidate species. Hence, each is assigned a PHSs with PSH-4 representing *C. cuarassu*, PSH-5 being *C. crioula*, PSH-8 being *C. tamoia* and PSH-10 corresponding to *C. pseudovaginuloides*. Regarding the rest of *Cratera* individuals included in this work, the two discovery methods coincide in assigning individuals F2789, MZUSP PL 1051, MZUSP PL 2154, MZUSP PL 2147, MZUSP PL 1014, MZUSP PL 2148 to one candidate species (PSH-1) and individuals MZUSP PL 0458 and MZUSP PL 0459 to another candidate species (PSH-9). Finally, in two cases a clade that mPTP considers as a single MOTU is divided into two candidate species by ABGD that are here designated as candidate species PSH-2, PSH-3, PSH-6 and
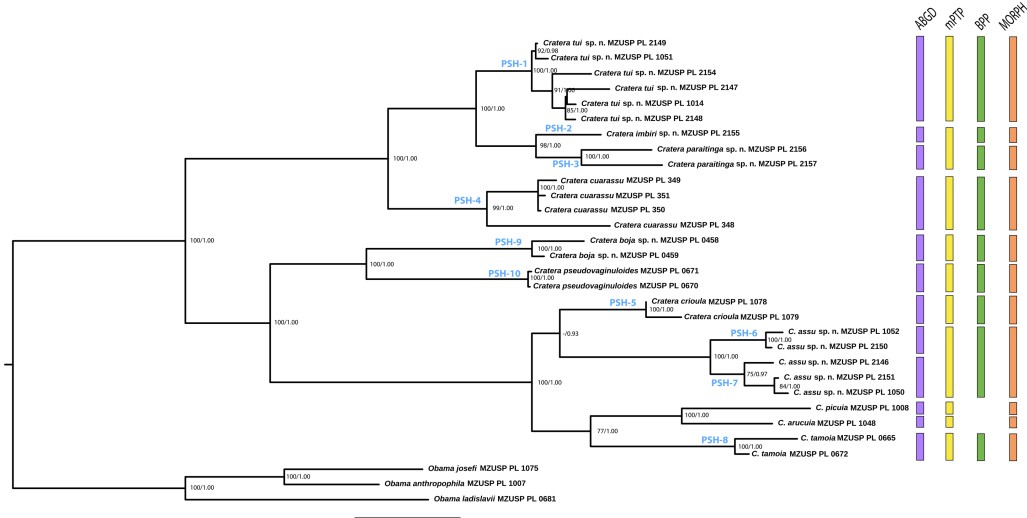

**Figure 2** **Phylogenetic tree showing the molecular results.** Phylogenetic tree showing the topology obtained by Bayesian Inference with the information of the concatenated six genes (18S rDNA + COI + 28S rDNA + Tnuc813 + EF + Nd4toCox1). Numbers at the nodes correspond to bootstrap values of the maximum likelihood analysis and those of Bayesian posterior probability (right). Correspondence with the Primary Species Hypothesis assigned by the molecular species delimitation methods is indicated at the nodes. Vertical bars to the right of the phylogeny show the summary of the species delimitation analyses by ABGD (purple), mPTP (yellow), BPP (green), and morphological (orange) approaches. Scale bar represents substitutions per site.

PSH-7. The species *C. picuia* and *C. arucuia* are not assigned to any PSH because they are singletons (constituted by a single individual) and thus are not included in the validation step. As a result, 10 PSHs have to be validated in BPP.

The species tree resulting from the *BEAST analysis (Fig. 3), used as the input tree to implement the BPP method, differs slightly from the tree inferred using the concatenated dataset. In the ML and BI phylogenies inferred from the concatenated dataset (Fig. 2), the clade formed by PSH-9 and PSH-10 is sister to the clade constituted by PSH-5, PSH-6, PSH-7 and PSH-8. However, in the *BEAST tree PSH-9 + PSH-10 form the sister clade of the rest of the species. These small differences could be due to the fact that *C. picuia* and *C. arucuia* were not included in the *BEAST analysis, but it does not affect the subsequent species assignment.

The different values of $\theta$ (ancestral population size) and $\tau$ (time of divergence) that are used in the 4 BPP analyses (M1, M2, M3 and M4) do not have a significant effect on the results of the BPP analyses (Fig. 3), except for the node that separates PSH-2 and PSH-3, the PP value of which is higher than 0.95 in the M2 and M4 models (small ancestral population size) and a little lower than 0.95 in the M1 and M3 models (large ancestral population size). Of the 10 PSHs that are analysed as starting hypotheses for BPP, only 9 are validated since PSH-6 and PSH-7 form a single SSH (SSH-6 in Fig. 3). In conclusion, the BPP results determine the presence of 9 SSHs, validating five new species of *Cratera* (SSH-1, SSH-2, SSH-3, SSH-6, and SSH-9) that will be described.
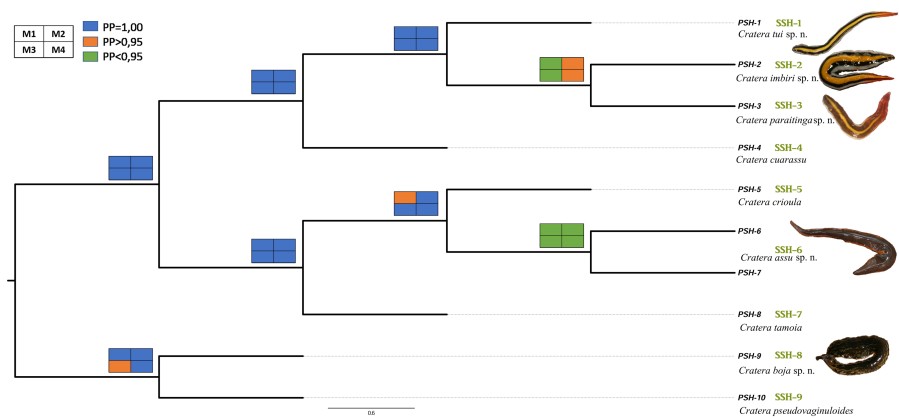

**Figure 3 Results of BPP molecular species delimitation.** Results of BPP represented in the species tree estimated with *BEAST for the genus *Cratera*. Colored squares at the nodes represent PP values for each of the applied models (M1, left up, M2, right up, M3, left down, M4, right down). Colors correspond to PP values: 1 (blue), >0.95 (orange), or <0.95 (green). Nomenclature of Primary Species Hypothesis (PSH) is shown at the terminals of the branches. On the right, green letters indicate the Secondary Species Hypothesis validated by BPP. Images of the worms correspond to the new species. Scale bar represents substitutions per site.

## Taxonomic account

**Order Tricladida Lang, 1884**
**Suborder Continenticola Carranza et al., 1998**
**Family Geoplanidae Stimpson, 1857**
**Genus *Cratera* Carbayo et al., 2013**
*Cratera assu* **Araujo, Carbayo, Riutort & Álvarez-Presas, sp. nov.**
urn:lsid:zoobank.org:act:DE3D812D-C387-40BD-9273-7BC1FE59D09C

**Synonymy.**
*Cratera* sp. 1: *Carbayo et al. (2013)*.
**Etymology.** The specific epithet *assu* means *large* in Tupi (indigenous Brazilian language; *Tibiriçá, 1984*). It refers to the large distal dilation of the ejaculatory duct. The specific epithet is invariable.
**Type locality.** Parque Nacional da Serra da Bocaina, São José do Barreiro, State of São Paulo, Brazil.
**Material examined.** All specimens collected in the Parque Nacional da Serra da Bocaina, São José do Barreiro, State of São Paulo, Brazil (−22.75, −44.62) by Carbayo and co-workers Holotype MZUSP PL 2150 (Field code F2825), sexually mature: 8 September 2008. Horizontal sections of pre-pharyngeal region-1 on 5 slides; transverse sections of pre-pharyngeal region-2 on 5 slides; sagittal sections of pharynx and copulatory apparatus on 34 slides. Paratypes: MZUSP PL 2146 (Field code F2025), sexually mature: 5 February 2008. Horizontal sections of pre-pharyngeal region-1 on 15 slides; transverse sections of pre-pharyngeal region-2 on 7 slides. Sagittal sections of copulatory apparatus on 15

slides. MZUSP PL 1050 (Field code F2807), sexually mature: 7 September 2008. Transverse sections of cephalic extremity on 7 slides; horizontal sections of ovaries on 5 slides; transverse sections of pre-pharyngeal region on 6 slides; sagittal sections of pharynx on 12 slides; sagittal sections of copulatory apparatus on 11 slides. MZUSP PL 1052 (Field code F2821), sexually mature: 8 September 2008. Sagittal sections of a piece of the body behind cephalic extremity on 71 slides; sagittal sections of pharynx and copulatory apparatus on 71 slides. MZUSP PL 2151 (Field code F2834): 8 September 2008. Preserved in 80% ethanol.

## Diagnosis

Species of *Cratera* 50–65 mm long (38–53 mm in preserved condition); dorsum chestnut brown to black brown, excepting the yellow orange or grayish body margins. A thin grayish median stripe may be present. Eyes dorsal. Pharynx bell-shaped. Penis papilla horizontal; distal portion of ejaculatory duct very widened, occupying most of the of the penis papilla. Common glandular ovovitelline duct absent.

## Description

Live animals ranging between 50–65 mm in length, and 3.0–3.5 mm in width ($n = 3$). Preserved, ranging between 38–53 mm and 5-7 mm, respectively ($n = 3$). Body elongate, slightly lanceolated, with maximum width at the level of the pharynx. Anterior to the pharynx, the body narrows gradually towards the rounded anterior tip; posterior to the pharynx, body becomes narrower abruptly near posterior, pointed tip. Dorsum slightly convex, ventral side flattened. Creeping sole as wide as 92–94% of body width at the pre-pharyngeal region. In holotype, mouth and gonopore at a distance from anterior extremity equal to 70–74% and 85% of body length, respectively ($n = 2$).

Color of the dorsum varies from chestnut brown (Fig. 4A) to black brown (Fig. 4B). A submarginal yellow orange or grayish stripe is present in anterior tenth of the body; posteriorly, this stripe becomes marginal. However, this stripe, measuring 8% of body width, is inconspicuous in some animals (Fig. 4B). A thin grayish median stripe, 4% of body width, may also occur (Fig. 4A). Ventral surface varies from sand yellow (Fig. 4A, inset) to light ivory (Fig. 4B, inset).

Each eye is formed by a single pigment cup 25–30 µm in diameter. Clear halos around eyes were not observed. Eyes contour anterior extremity of the body in a single row along the anterior 13–16% of body length ($n = 2$); then they spread onto the dorsum in a lateral band with 1/3th of the body width on each side, reaching the posterior tip.

Sensory pits, 13–18 µm deep, in a uniserial ventro-lateral row, from very anterior tip extending posteriorly up to 29% of body length. Dorsal and ventral epidermis in pre-pharyngeal region pierced by necks of two types of gland cells producing erythrophil and cyanophil granules. Rhabditogen cells discharge their content through dorsal epithelium. Glandular margin constituted by two types of gland cells, one abundant type producing xanthophil granules and a less abundant type secreting erythrophil granules.

Cutaneous musculature constituted of a subepidermal circular layer, followed by two diagonal layers with decussate fibers, and then a strongly developed longitudinal layer with fibers arranged in bundles (Fig. 4C). Longitudinal layer thickness ranging between 40–62.5

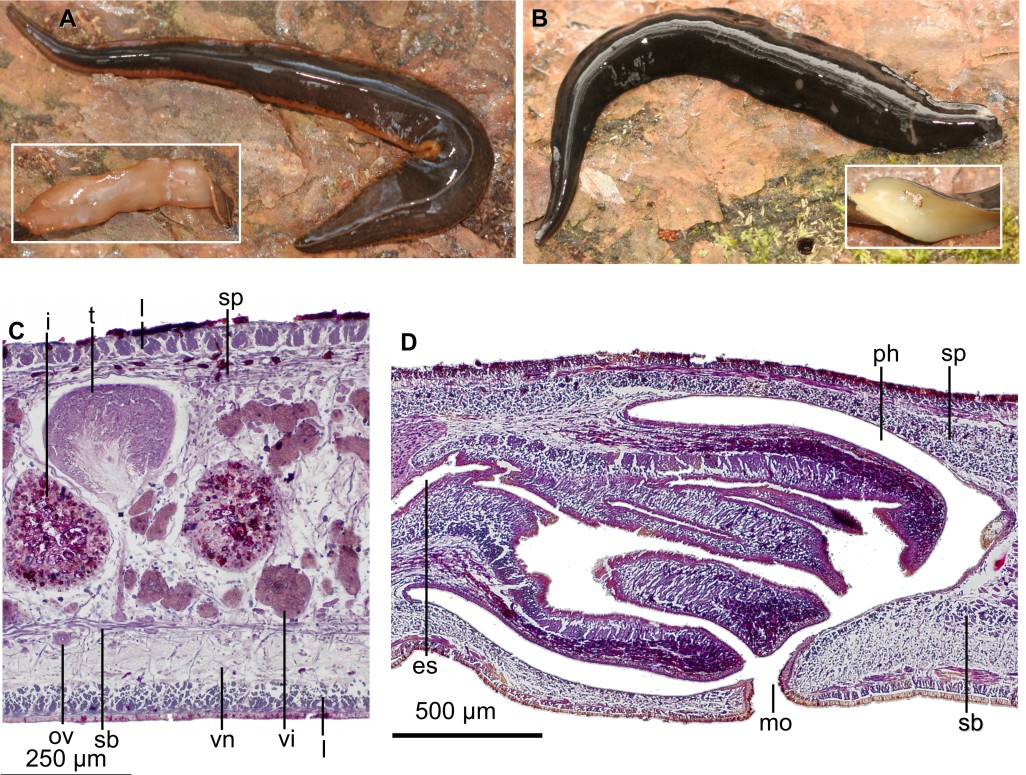

**Figure 4** **_Cratera assu_ sp. n. morphological characters.** _Cratera assu_ sp. n. (A) Dorsal and ventral (inset) views of living paratype MZUSP PL 1050. Scale bar not available. (B) Dorsal and ventral (inset) views of living paratype MZUSP PL 1052. Scale bar not available. (C) Photomicrograph of a sagittal section of the pharynx of holotype. (D) Photomicrograph of a transverse section of pre-pharyngeal region of paratype MZUSP PL 1050.

μm ($n=4$) dorsally, and 40–90 μm ventrally; dorsally, fibers gathered into well-delimited and more compact bundles than ventrally. Thickness of cutaneous muscle coat ranging between 8.9%–13.4% ($n=4$).

Three parenchymal muscle layers throughout the body: a dorsal layer of diagonal decussate fibers (20–21 μm thick; $n=2$), a transverse supraintestinal layer (145–157 μm, $n=2$), and a transverse subintestinal one (90 μm; $n=2$). Central nervous system as a ventral nerve plate. Cerebral ganglia were not discerned.

Mouth located shortly behind middle of pharyngeal pouch. Pharynx bell-shaped, with dorsal insertion posterior to the ventral one about 40% of pharyngeal length ($n=2$; Fig. 4D). Esophagus length, 33% of pharyngeal length ($n=2$). Outer pharyngeal epithelium underlain by a one-fiber-thick layer of longitudinal muscle, followed by a circular one (6–10 μm thick; $n=3$); inner epithelium underlain by a layer (70–150 μm; $n=3$) of circular fibers with interspersed longitudinal fibers, followed by a longitudinal muscle layer (9–10 μm; $n=3$). Pharyngeal pouch 1.6–2.6 mm anterior to prostatic vesicle ($n=2$).

Testes dorsal, located under the supraintestinal transverse muscle layer, partially placed between the intestinal diverticula (Fig. 4C). Sperm ducts run immediately above the

subintestinal muscle layer, dorso-medially to the ovovitelline ducts (Fig. 4C). Each sperm duct communicates with the respective short diverticulum projected from the proximal, slightly widened section of the prostatic vesicle. The prostatic vesicle exhibits an inverted U-shaped loop before penetrating the penis bulb and communicating with the ejaculatory duct (Fig. 5A). Prostatic vesicle lined with a ciliated, tall epithelium, the free surface of it is irregular. This epithelium is traversed by cells producing fine erythrophil granules. Prostatic vesicle surrounded by a 15–33-μm thick ($n = 4$) circular muscle layer. Proximal portion of the ejaculatory duct slightly sinuous. At the basis of the penis papilla, ejaculatory duct widens to form a very large cavity (Figs. 5A–5C). At the tip of penis papilla this cavity narrows to open into male atrium. Ejaculatory duct lined with a cuboidal, ciliated epithelium and surrounded by a 10–25–μm thick circular muscle, followed by an additional, one-fiber-thick longitudinal muscle ($n = 3$).

Penis papilla a long, more or less conical, protrusible organ extending along the length of the male atrium or even beyond (Figs. 5A–5C). Lining epithelium of the widened portion of the ejaculatory duct cuboidal, not ciliated, pierced by necks of cell producing minute erythrophil granules (<0.2 μm), and underlain by a layer (20–28 μm; $n = 2$) of circular muscle, followed by a simple layer of longitudinal muscle. The penis papilla is clothed with a cuboidal, non-ciliated epithelium, which is traversed by necks of cells producing erythrophil granules and is underlain by a 20-μm-thick layer of circular muscle ($n = 3$) with some interspersed longitudinal fibers.

Male atrium ample and smooth, not narrowed distally. It is lined with a cuboidal, non-ciliated epithelium, which is pierced by necks of cells producing granules with weak xanthophil affinity. Male atrial epithelium underlain by a muscle layer (18–45 μm; $n = 3$) of circular fibers.

Vitellaria well developed. Ovaries mature, more or less ovoid, with maximum length ranging between 400–440 μm antero-posteriorly ($n = 2$). They are located immediately above the ventral nerve plate, at a distance from anterior tip equivalent to 21% of body length. Ovovitelline ducts arise from external side of the ovaries, whereafter they run backwards above ventral nerve plate. The ducts ascend laterally to the female atrium, posteriorly and medially inclined, to unite dorsally to the atrium (Fig. 5A). Half of the ascending portion of these ducts receives secretion of shell glands (Fig. 5A). Ovovitelline ducts open into the female genital canal. Common glandular ovovitelline duct absent. The female canal is a long forwards directed canalicular projection of the posterior portion of the female atrium (Fig. 5D). Female atrium, as long as male atrium, and funnel-shaped. Lateral walls of female atrium partially occlude its lumen. Its lining epithelium is columnar with the free surfaces of the cells being erythrophil and provided with small recesses (Fig. 5D).

In the anterior portion of female atrium, the epithelium is 25–40 μm high ($n = 3$); posteriorly it becomes taller and its nuclei arranged at different heights within the cells, thus giving the epithelium a multilayered aspect. Necks of cells producing erythrophil granules pierce this epithelium, which is underlain by two layers of muscles; the anterior half of the atrium is surrounded by a 5-μm-thick circular muscle followed by a 3-μm-thick longitudinal one ($n = 3$). Arrangement of these layers is inverted on the posterior half of

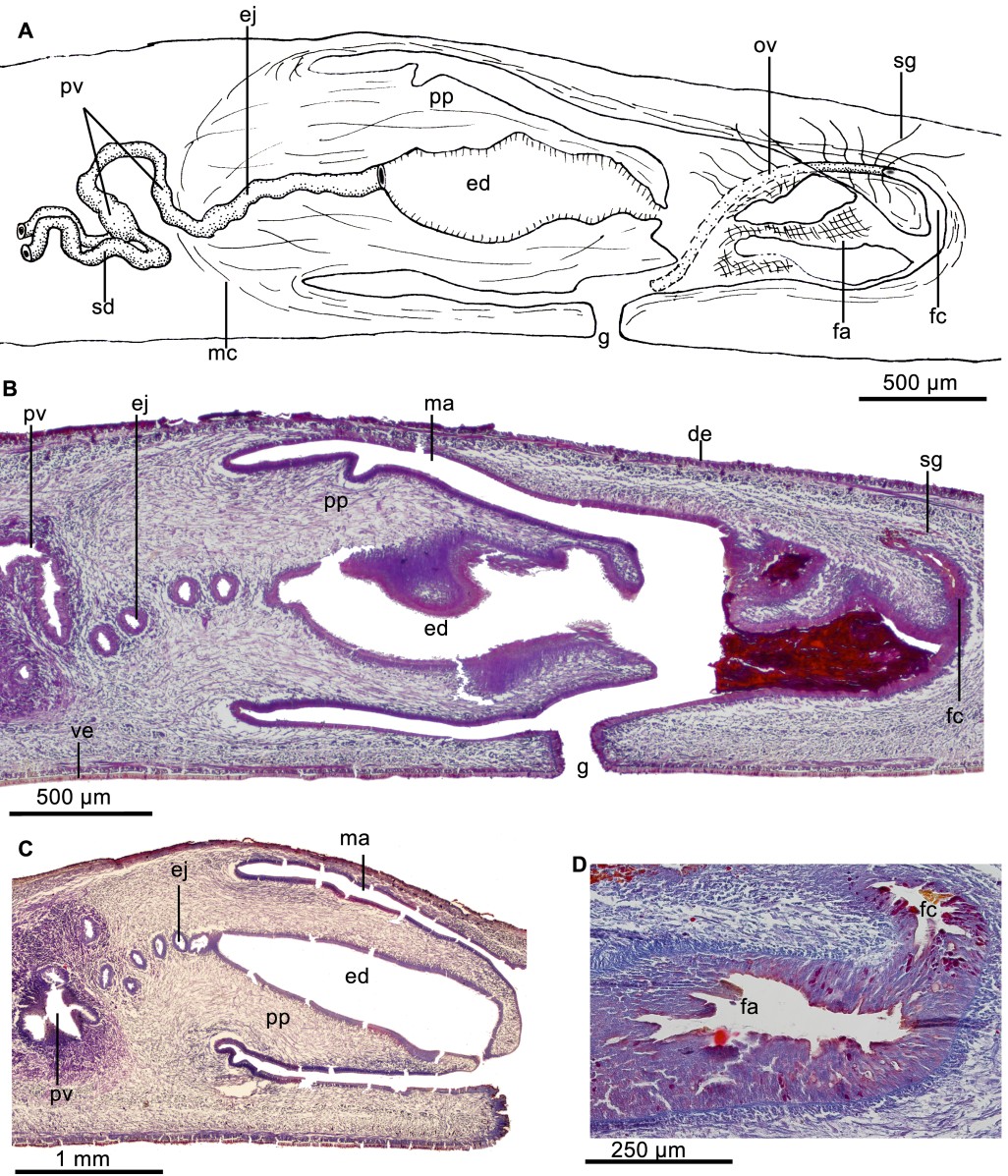

**Figure 5** *Cratera assu* **sp. n. morphological details.** *Cratera assu* sp. n. (A) Diagrammatic representation of the copulatory apparatus of holotype from sagittal sections. (B) Photomicrograph of a sagittal section of the copulatory apparatus of holotype. (C) Photomicrograph of a sagittal section of the copulatory apparatus of paratype MZUSP PL 1052. (D) Photomicrograph of a sagittal section of the female atrium of paratype MZUSP PL 2146.

the female atrium, i. e., a layer (5–7 µm) of longitudinal muscles is followed by a layer (12–20 µm) of circular fibers ($n = 3$).

## Discussion

Regarding the external aspect, *C. cryptolineata* *Rossi & Leal-Zanchet, 2017* is the only species of the genus similar to *C. assu* in that both species display a uniform dark dorsum with clear

margins. However, their copulatory apparatuses are notably different from each other in that in *C. assu* the distal, widened portion of the ejaculatory duct is much larger, occupying most of the penis papilla. This remarkable trait is also shared with *C. cuarassu*, but in this latter species the penis papilla is vertically oriented (vs. horizontally in *C. assu*). The molecular delimitation methods all clearly point to *C. assu* being a species differentiated from the rest of species molecularly analysed in the present study. On the other hand, although the primary hypothesis pointed that this lineage could be divided into two species (ABGD analysis), neither the molecular validation nor the morphological analysis found evidence of this possibility.

### *Cratera boja* Araujo, Carbayo, Riutort & Álvarez-Presas, sp. nov.
urn:lsid:zoobank.org:act:46D76DD7-B129-461D-8803-3E918AA4601C

**Etymology.** The specific epithet *boja* means *intermediate, middle* in Tupi (*Tibiriçá, 1984*). It refers to intermediate size of the distal dilation of the ejaculatory duct. The specific epithet is invariable.

**Type locality.** Parque Nacional da Serra da Bocaina, São José do Barreiro, State of São Paulo, Brazil.

**Material examined.** All specimens collected in Parque Nacional da Serra da Bocaina, São José do Barreiro, State of São Paulo, Brazil (−22.75, −44.62), in September 2008 by Carbayo and co-workers. Holotype MZUSP PL 0459 (Field code F2829), sexually mature: transverse sections of cephalic extremity on 4 slides; horizontal sections of a portion of the body behind cephalic extremity on 4 slides; transverse sections of pre-pharyngeal portion on 7 slides; sagittal sections of pharynx on 6 slides; sagittal sections of copulatory apparatus on 5 slides (of which one slide was lost). MZUSP PL 0458 (Field code F2828), incompletely mature: horizontal sections of a portion behind anterior extremity on 6 slides; transverse sections of the pre-pharyngeal portion on 22 slides; sagittal sections of pharynx on 13 slides; sagittal sections of copulatory apparatus on 14 slides.

## Diagnosis

Species of *Cratera* 34 mm long preserved; dorsum olive grey with large black patches, being more concentrated in paramedian bands; eyes marginal; pharynx bell-shaped; distal portion of ejaculatory duct widened to occupy half of the penis papilla; male atrium separated from female atrium by a narrowing; penis papilla shorter than male atrium; prostatic vesicle with inverted-U shape in lateral view; penis papilla postero-dorsally oriented; numerous cyanophil cell necks piercing roof of male atrium; female atrium half the length of the male atrium; common glandular ovovitelline duct absent.

## Description

Preserved holotype 34 mm in length and four mm in width. Paratype was measured only in its width, 4.5 mm. Body elongate, with nearly parallel margins along most of body length. Anterior extremity rounded; posterior pointed. Dorsal side convex; ventral side flattened.

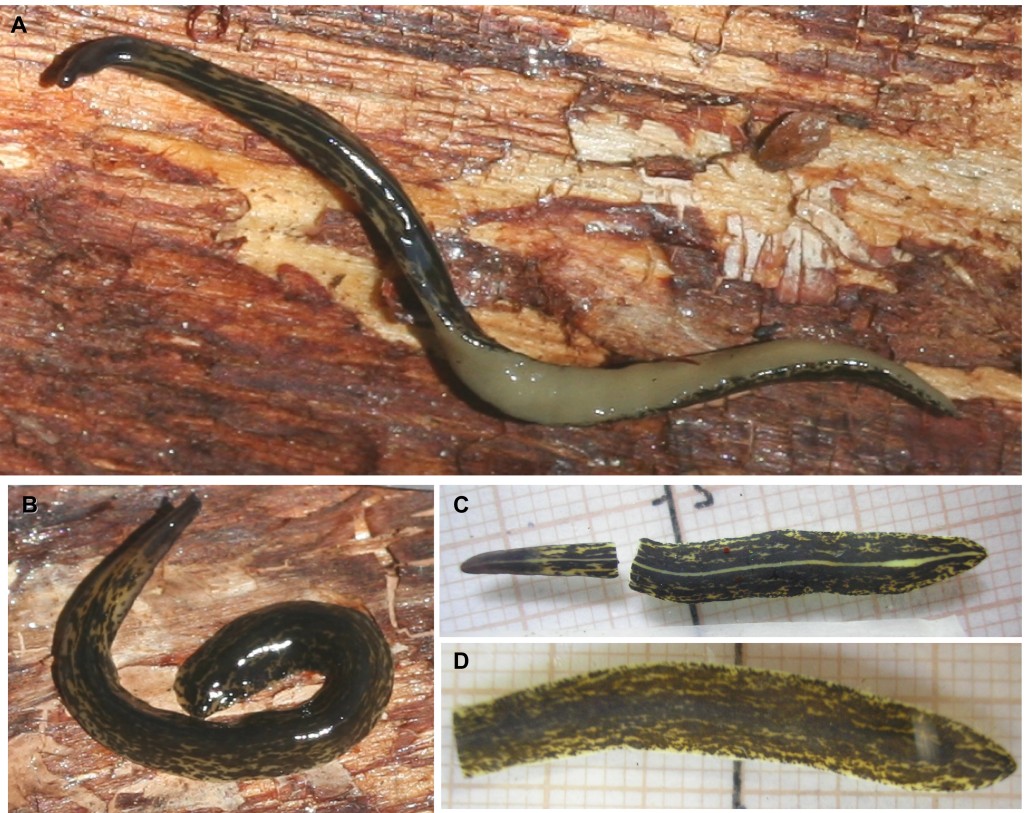

**Figure 6** *Cratera boja* **sp. n. morphological details.** *Cratera boja* sp. n. (A) Living holotype. Scale bar not available. (B) Dorsal view of living paratype. Scale bar not available. (C) Dorsal view of holotype preserved on millimeter graph paper after cutting off a piece of the body. (D) Dorsal view of the paratype, preserved on millimeter graph paper after cutting off anterior extremity of the body.

Creeping sole, 92–95% of body width at the pre-pharyngeal region. Mouth at a distance from anterior extremity equal to 71% of body length; gonopore at 87% (holotype).

Dorsum spotted black on olive grey ground color (Figs. 6A–6D). Large black patches join each other. Patches distributed all over dorsum and especially concentrated in a median band (33% of body width). This band may be divided by a thin midline, measuring 6% of body width (Fig. 6C). Ventral surface olive gray (Fig. 6A), gray at anterior extremity.

Each eye is formed by a single pigmented cup 32–40 μm in diameter. Clear halos around eyes absent. Eyes contour anterior extremity in a single row, posteriorly eye cups are arranged marginally in 1–3 rows up to the posterior tip of the body.

Sensory pits were not found despite intensive search on the sections, albeit that these are partially damaged. Dorsal and ventral epidermis in pre-pharyngeal region pierced by necks of two types of gland cells producing erythrophil and cyanophil granules, respectively. Furthermore, rhabditogen cells discharge their content through dorsal epithelium and, much more rarely, through the ventral epithelium as well. Glandular margin not well defined (Fig. 7A), constituted by three types of gland cells, one abundant

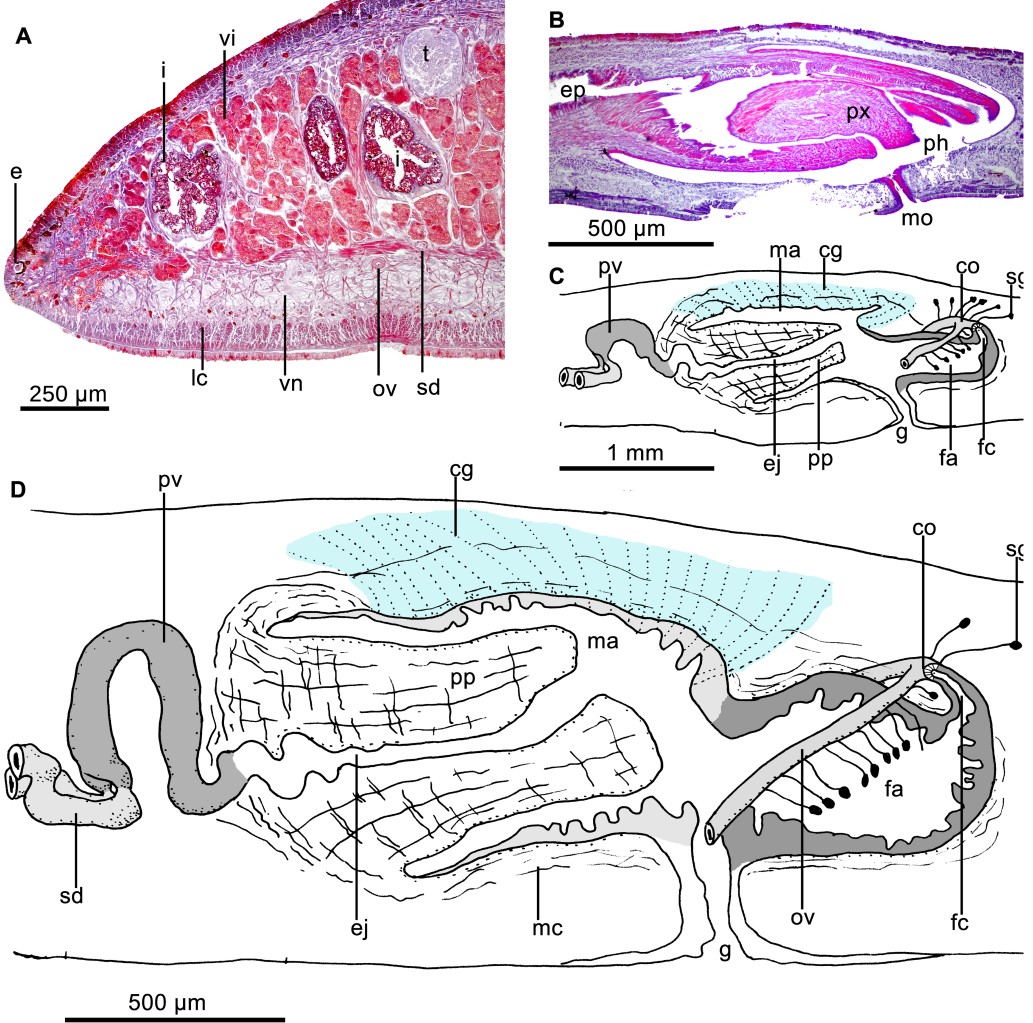

**Figure 7** *Cratera boja* **sp. n. morphological details.** *Cratera boja* sp. n. (A) Photomicrograph of a transverse section of pre-pharyngeal region of holotype. (B) Photomicrograph of a sagittal section of the pharynx of paratype. (C) Diagrammatic representation of the copulatory apparatus of paratype from sagittal sections. (D) Diagrammatic representation of the copulatory apparatus of holotype from sagittal sections.

type producing xanthophil granules and a less abundant type secreting erythrophil granules, and rhabditogen cells.

Cutaneous musculature constituted of a subepidermal circular layer, followed by two diagonal layers with decussate fibers, and an inner, strongly developed longitudinal layer with fibers arranged in bundles (Fig. 7A). Longitudinal layer 55 µm thick dorsally and 90 µm ventrally. Dorsal fibers of this layer are gathered into well-delimited and more compact bundles than ventrally. Thickness of cutaneous muscle coat, 12.5% of body height (holotype).

Three parenchymal muscle layers throughout the body: a dorsal layer of diagonal decussate fibers (20 µm thick, holotype), a transverse supraintestinal layer (60 µm), and

a transverse subintestinal one (70 μm). Central nervous system as a ventral nerve plate. Cerebral ganglia more or less five mm long, starting at 0.5 mm from anterior extremity (1.5% of body length, holotype).

Mouth located at the end of second third of pharyngeal pouch (Fig. 7B). Pharynx bell-shaped, with dorsal insertion posterior to the ventral one at the equivalent of 36% of pharyngeal length ($n = 2$). Esophagus length, 21% of pharyngeal length ($n = 2$). Outer pharyngeal epithelium underlain by a one-fiber-thick longitudinal muscle layer followed by a circular one (4 μm thick; $n = 2$); inner epithelium underlain by a circular muscle layer (22–60 μm thick; $n = 2$), followed by a longitudinal (5 μm; $n = 2$). Pharyngeal pouch of holotype 2.75 mm anterior to prostatic vesicle.

Testes dorsal, located under the supraintestinal transverse muscle layer, partially placed between the intestinal diverticula (Fig. 7A). Posteriormost testes lateral to ventral insertion of the pharynx. Sperm ducts run immediately above the subintestinal muscle layer, dorso-medially to the ovovitelline ducts (Fig. 7A). Below prostatic vesicle, sperm ducts curve medially and anteriorly, to communicate separately with the respective short lateral diverticula of the extrabulbar and tubular prostatic vesicle (Figs. 7C, 7D). In lateral view, the latter has the shape of an inverted U. Prostatic vesicle approaches anterior region of penis bulb to communicate with ejaculatory duct.

Prostatic vesicle lined with a ciliated, columnar epithelium, traversed by necks of two types of gland cells producing erythrophil and cyanophil granules, respectively; the cyanophil glands are much more abundant (Fig. 8A). Prostatic vesicle surrounded by a 130-μm-thick circular muscle layer. Proximal portion of ejaculatory duct sinuous; distal portion straight, traversing center of penis papilla; midway of penis papilla, ejaculatory duct widens to give rise to a relatively large, funnel-shaped cavity, which opens at the tip of the penis papilla (Figs. 7C–7D, 8A). Ejaculatory duct lined with a cuboidal, ciliated epithelium, which at its dilated portion is pierced by necks of cells producing xanthophil granules. Ejaculatory duct surrounded by a 20-μm-thick circular muscle layer.

Penis papilla more or less conical, 1.3–1.5 times longer than its diameter and as long as the male atrium or shorter (Figs. 7C–7D, 8A). Penis papilla clothed with non-ciliated epithelium. Columnar epithelium lines basal half of papilla; cuboidal epithelium lines distal half. Epithelium of papilla pierced by necks of three types of cells producing erythrophil, xanthophil and cyanophil granules, respectively (but the latter appears erythrophil in paratype); necks of cyanophil glands only piercing epithelium of basal half of penis papilla. Epithelium of penis papilla underlain by a 8–10-μm-thick layer of circular muscle fibers, followed by a longitudinal layer, 10μm thick. Parenchyma of penis papilla richly traversed by intermingled circular and longitudinal muscle fibers.

Male atrium ample and smooth, lined with a columnar, non-ciliated epithelium. Epithelium clothing roof of male atrium pierced by very numerous necks of cells producing cyanophil granules. Ventral portion of male atrium pierced by less numerous necks of glands producing erythrophil granules. Atrial epithelium underlain by a subepithelial layer (10–12 μm thick) of circular muscle fibers, followed under dorsal epithelium by a 3–5–μm-thick longitudinal muscle layer.

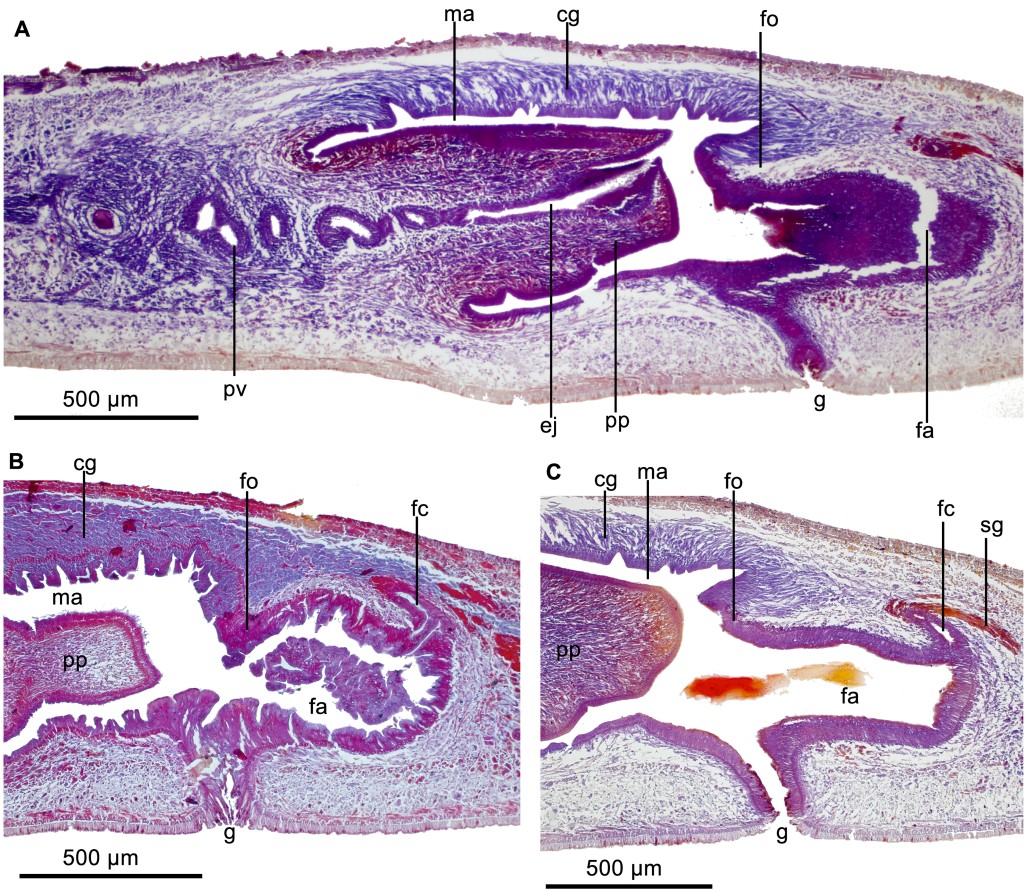

**Figure 8** *Cratera boja* sp. n. morphological features. *Cratera boja* sp. n. Photomicrographs of sagittal sections. (A) Copulatory apparatus of paratype. (B) Female atrium of holotype. (C) Female atrium of paratype.

Vitellaria well developed. Ovaries were not found in histological sections and were probably destroyed in the portion from which a tissue sample was extracted. Ovovitelline ducts well developed, having their lumen well defined. They ascend laterally to proximal portion of the female atrium, posteriorly and medially inclined. Subsequently, the ducts unite dorsally to the posterior portion of atrium. Distal half of ascending portion of these ducts receives shell glands. Ovovitelline ducts open into female genital canal, i.e., a common glandular ovovitelline duct being absent (see 'co' in Figs. 7C–7D). Female genital canal projects forwards from the postero-dorsal portion of the female atrium. Female atrium ample, 55–57% the length of male atrium and lined with 30–70 μm high, non-ciliated epithelium, with stratified aspect (Figs. 8A–8C). Surface of epithelium sinuous; subapical portion of epithelial cells cyanophil (Fig. 8B) or erythrophil (Fig. 8C). Necks of two types of cells, producing erythrophil and cyanophil granules, respectively, pierce the atrial epithelium, which is underlain by a 18–26–μm-thick and dense layer of circular muscle, followed by a 3–4–μm-thick longitudinal muscle layer.

## Discussion

*C. boja* is distinctive among all species of the genus in the spotted color pattern of the dorsal side. Regarding its copulatory apparatus, there are five species with a similar aspect of the copulatory apparatus and size of the dilation of the ejaculatory duct, namely, *C. aureomaculata* *Rossi & Leal-Zanchet, 2017*, *C. nigrimarginata* *Rossi & Leal-Zanchet, 2017*, *C. pseudovaginuloides*, *C. tamoia*, and *C. viridimaculata* *Negrete & Brusa, 2016*. However, (a) all of these five species have a common glandular ovovitelline duct (absent in *C. boja*; but see redescription of *C. pseudovaginuloides* in *Riester, 1938*); (b) the male and female atria are not separated by a constriction in *C. pseudovaginuloides* (separated in *C. boja*); (c) the penis papilla is longer than the male atrium in *C. nigrimarginata* and *C. pseudovaginuloides* (shorter in *C. boja*); (d) the prostatic vesicle is horizontal in *C. nigrimarginata* (inverted-U shaped in *C. assu*); (e) the female atrium is funnel-shaped, and its posterior section oriented upwards in *C. viridimaculata* (horizontal, and not funnel-shaped in *C. boja*); (f) the prostatic vesicle runs postero-dorsally in *C. aureomaculata* and *C. tamoia* (inverted U-shaped in *C. boja*); (g) the penis papilla exhibits a postero-ventral orientation in *C. tamoia* (postero-dorsal in *C. boja*); and (h) there is no accumulation of necks of cyanophil cells piercing the roof of the male atrium in *C. nigrimarginata*, *C. pseudovaginuloides*, and *C. tamoia* (cell necks present in *C. boja*).

The phylogenetic tree as well as the molecular species delimitation analyses also show that this species is distinct and well-delimited from *C. pseudovaginuloides* and *C. tamoia*, the two species included in these analyses that show similarities with *C. boja*.

### *Cratera tui* Araujo, Carbayo, Riutort & Álvarez-Presas, sp. nov.
urn:lsid:zoobank.org:act:FB96BE96-86AD-41FB-961F-C0337C56A596

**Synonymy.** *Cratera* sp. 4: *Carbayo et al. (2013)*.
**Etymology.** The specific epithet *tui* means *tiny, insignificant* in Tupi (*Tibiriçá, 1984*). It refers to the small distal dilation of the ejaculatory duct. The specific epithet is invariable.
**Type locality.** Parque Nacional da Serra da Bocaina, São José do Barreiro, State of São Paulo, Brazil.
**Distribution.** Parque Nacional da Serra da Bocaina, São José do Barreiro, State of São Paulo; Parque Nacional Itatiaia, Resende, State of Rio de Janeiro, Brazil.
**Material examined.** Holotype MZUSP PL 1051 (Field code F2809), sexually mature: Parque Nacional da Serra da Bocaina, São José do Barreiro, State of São Paulo (−22.75, −44.62). coll. F. Carbayo and co-workers, 7 September 2008; transverse sections of cephalic extremity on 4 slides; sagittal sections of a portion immediately behind cephalic extremity on 19 slides; horizontal sections of a immediately behind on 12 slides; transverse sections of pre-pharyngeal on 16 slides; transverse sections of pharynx on 17 slides; sagittal sections of copulatory apparatus on 23 slides. Paratypes: MZUSP PL 1014 (Field code F2031), incompletely mature: Ibidem, 7 September 2008; sagittal sections of copulatory apparatus on 6 slides. MZUSP PL 2148 (Field code F2054), incompletely mature: Ibidem, 9 September 2008, sagittal sections of pharynx and copulatory apparatus on 13 slides (of

which 3 slides were lost). MZUSP PL 2147 (Field code F2040), juvenile: Ibidem, 9 February 2007, preserved in 80% ethanol. MZUSP PL 2149 (Field code F2798): Ibidem, 7 September 2008; preserved in 80% ethanol. MZUSP PL 2154 (Field code F5178), incompletely mature: Parque Nacional de Itatiaia, Resende, State of Rio de Janeiro (−22.43328, −44.61539), Brazil, coll. F. Carbayo and co-workers, 5 April 2012, horizontal sections of a body portion behind cephalic extremity on 7 slides; transverse sections of pre-pharyngeal region on 15 slides; sagittal sections of pharynx and copulatory apparatus on 30 slides.

## Diagnosis

Species of *Cratera* 45–70 mm long in preserved condition; dorsum with a melon yellow median stripe, bordered on either side by a jet-black stripe external to which a marginal traffic white stripe; body margins jet black. Anterior 1/5th of the body colored with a gradient of carmine red; eyes marginal; pharynx cylindrical; pharyngeal pouch 0.6 mm anterior to the prostatic vesicle (equal to 1% of body length); penis papilla shorter than male atrium; distal dilatation of ejaculatory duct relatively small; female atrium 2.5 times longer than the male atrium; common glandular ovovitelline duct long.

## Description

The longest and only mature animal (holotype) measured 58 mm in length, and 7 mm in width. Body slightly lanceolate, with maximum width at the level of the pharynx. Anterior to the pharynx, narrows gradually towards the rounded tip; posterior to the level of the pharynx, body narrows abruptly close to posterior tip. Dorsum slightly convex, ventral side flattened. Creeping sole 90–95% of body width at pre-pharyngeal region ($n = 2$). Mouth at a distance from anterior extremity equal to 73% of body length; the gonopore at 82% (holotype).

Dorsum with a melon yellow median stripe, 28% of body width, which is bordered on either side by a jet black stripe, 22% of the body width (Figs. 9A–9B). External to jet black stripes, a traffic white marginal stripe, 10% of body width; body margin (3% of the body width) jet black. Anterior 1/5th of the body colored with a gradient of carmine red, dorsally and ventrally; otherwise, ventral side grey white (Fig. 9B).

Each eye is formed by a single pigmented cup with 33–35 μm in diameter. No clear halos around eyes. Eyes contour the anterior extremity in a single row and run along body margin until posterior tip.

Sensory pits, 19–20 μm deep, as a uniserial ventro-lateral row, from anterior extremity extending backwards to at least 9% of body length (holotype). Rhabditogen cells, and necks of two types of gland cells, producing xanthophil and erythrophil granules, respectively, pierce pre-pharyngeal dorsal epithelium; the latter gland cells also pierce ventral epithelium. Conspicuous glandular margin constituted by abundant glands producing erythrophil granules (Fig. 10A).

Cutaneous musculature consisting of three layers: a subepidermal circular layer, followed by two diagonals with decussate fibers, and a subjacent longitudinal one, the latter 30–40 μm thick dorsally and 40–65 μm thick ventrally ($n = 2$). Fibers of the longitudinal layer gathered in bundles, which are dorsally better delimited than ventrally. Relative thickness of cutaneous muscle coat, 9.1–12.7% ($n = 2$).

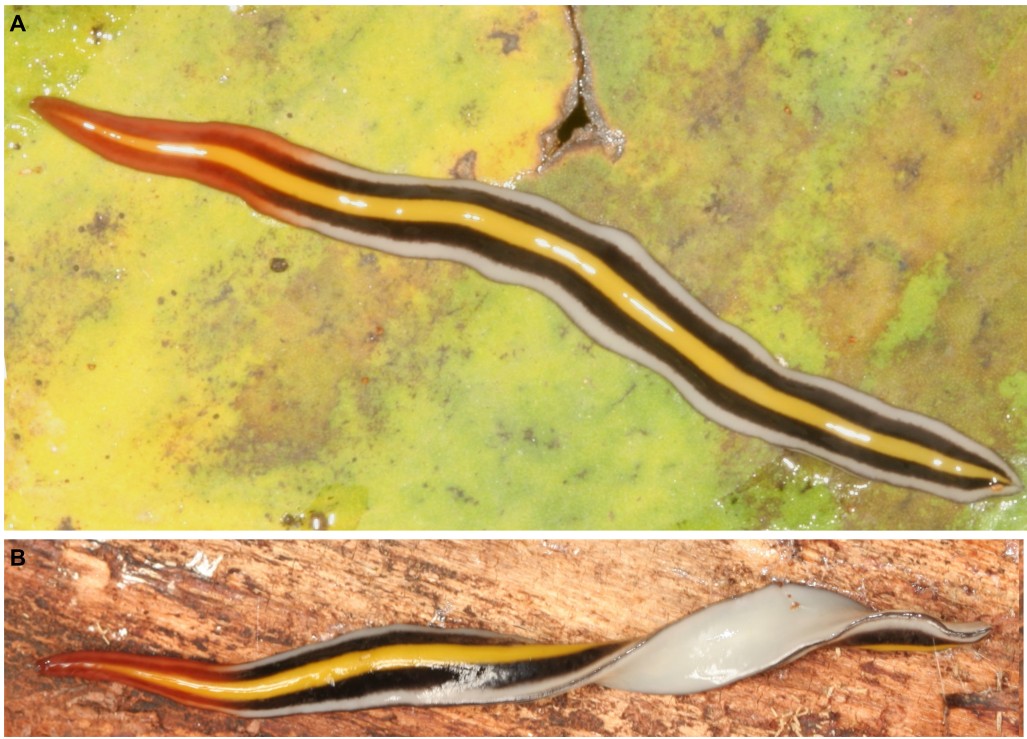

**Figure 9** *Cratera tui* **sp. n. morphological features.** *Cratera tui* sp. n. (A) Dorsal view of living paratype MZUSP PL 2154. (B) Dorsal view of living holotype partially twisted. Scale bars not available.

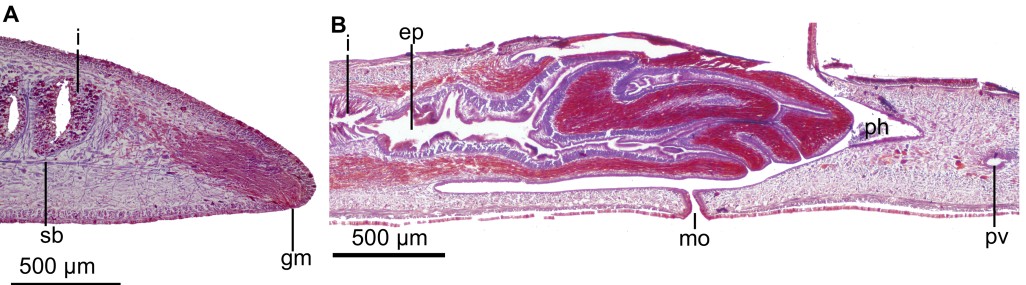

**Figure 10** *Cratera tui* **sp. n. morphological features.** *Cratera tui* sp. n. (A) Photomicrograph of a transverse section of pre-pharyngeal region of holotype. (B) Diagrammatic representation of the pharynx of paratype MZUSP PL 2154 from sagittal sections.

Three parenchymal muscle layers throughout the body, all constituted by fibers relatively densely packed: a dorsal layer of diagonal decussate fibers (15 μm thick, holotype), a transverse supraintestinal layer (25 μm), and a transverse subintestinal one (25 μm). In paratype MZUSP PL 2154, these layers measured, 20 μm, 72 μm, and 63 μm, respectively.

Central nervous system as a ventral nerve plate. Clearly evident cerebral ganglia were not found.

Mouth in middle of pharyngeal pouch ($n = 2$) (Fig. 10B). Pharynx cylindrical, with dorsal insertion posterior to the ventral one at the equivalent of 20% of pharyngeal length ($n = 2$). Esophagus length, ranging between 10–20% of pharyngeal length. Outer pharyngeal epithelium underlain by a one-fiber-thick layer (3–4 μm) of longitudinal muscle fibers, followed by a layer of circular fibers (7–8 μm); inner pharyngeal epithelium underlain by a layer (80–100 μm) of circular fibers, followed by a layer (10 μm) of longitudinal fibers ($n = 2$). Pharyngeal pouch 0.2–1.0 mm anterior to the prostatic vesicle ($n = 4$; 0.75 mm in holotype).

Testes mature, dorsal, located under supraintestinal transverse muscle layer, partially placed between the intestinal diverticula. The testes extend from level of ovaries to nearly root of pharynx. Sperm ducts communicate with the respective lateral branches of the prostatic vesicle (Fig. 11A). Paired portion of the prostatic vesicle represents ca. half of the total length of this vesicle. Extrabulbar prostatic vesicle more or less pear-shaped in lateral view, with posterior portion running posteriorly and upwards until anterior region of penis bulb. Vesicle lined with a columnar, ciliated epithelium, which is pierced by necks of cells producing fine erythrophil granules. A 20-μm-thick layer of circular muscle fibers surrounds vesicle. Inside penis papilla vesicle communicates with a horizontal, initially sinuous ejaculatory duct which is lined with a cuboidal, ciliated epithelium. Ejaculatory duct surrounded by a 5-μm-thick layer of circular muscle fibers. Near tip of penis papilla, lumen of the ejaculatory duct doubles its width to form a small cavity (Figs. 11A–11C).

Penis papilla short, conical and blunt, horizontally oriented; shorter than male atrium (Figs. 11A–11B). Male atrium slightly folded in its proximal region. Penis papilla and male atrium clothed with a cuboidal non-ciliated epithelium, pierced by necks of two types of cells, producing erythrophil and xanthophil granules, respectively. Epithelium of penis papilla and that of male atrium underlain by a 7-μm-thick layer of circular muscle fibers, followed by a 7-μm-thick layer of longitudinal muscle fibers.

Vitellaria well developed. Ovaries elongate, more or less ovoid, measuring 250 μm in antero-posterior direction. They are located immediately above the ventral nerve plate, at a distance from anterior tip equivalent to 30% of body length. Ovovitelline ducts arise from dorso-lateral side of ovaries and run backwards above the ventral nerve plate. They ascend laterally to female atrium to unite dorsally to female atrium to form common glandular ovovitelline duct (Fig. 11A). Distal ascending portion of oviducts receiving shell glands. Common glandular ovovitelline duct long (1.2 mm, i.e., 1/3th of the length of female atrium in holotype), communicating with female genital canal, the latter being a projection of the posterior portion of female atrium directed forwards and dorsally. Female atrium funnel-shaped, with a length 2.5 times that of male atrium. Lateral wall of female atrium with folds narrowing its lumen. Female atrium lined with a non-ciliated, 25-μm-tall epithelium along anterior 4/5th of its length. Posterior 1/5th lined with a 50-μm-high epithelium that might display a multilayered aspect; quality of the sections precluded confirmation. Necks of cells producing erythrophil granules pierce female epithelium, which is underlain by a layer of circular muscle fibers (7-μm-thick, holotype), followed by a layer of longitudinal fibers (7-μm-thick).

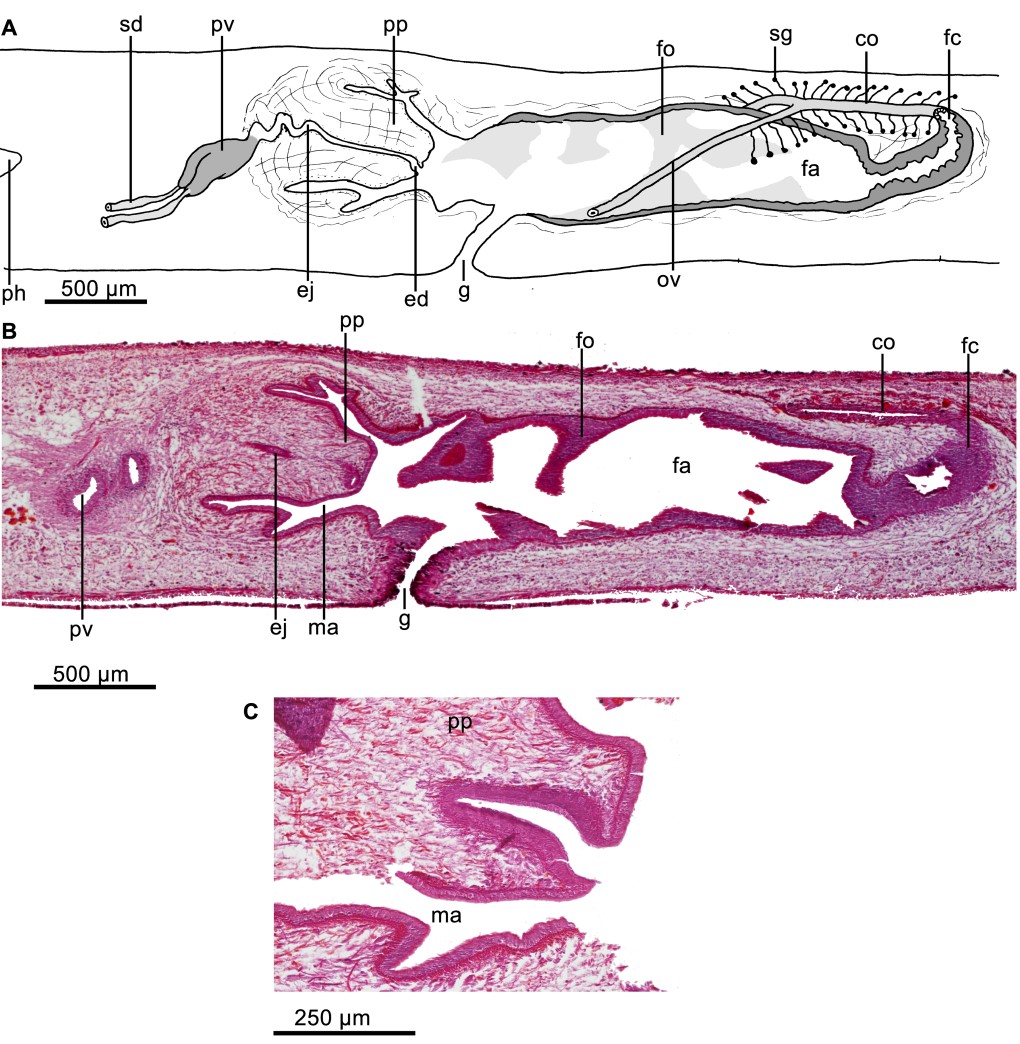

**Figure 11 Cratera tui sp. n. morphological details.** *Cratera tui* sp. n. Holotype. (A) Diagrammatic representation of the copulatory apparatus from sagittal sections. (B) Photomicrograph of a sagittal section of copulatory apparatus. (C) Photomicrograph of a sagittal section penis papilla.

## Discussion

Among all species of the genus, only *C. taxiarcha* (*Marcus, 1951*) resembles *C. tui* in the three-color striped pattern of the dorsum, composed of white, yellow, and black colors. However, in *C. taxiarcha*, the median stripe is white (vs. yellowish in *C. tui*). Regarding the copulatory apparatus, all species in the genus possess a female atrium as long as the male atrium, with minor variations, whereas in *C. tui* it is 2.5 times longer. The molecular delimitation methods all clearly point to *C. tui* being a species differentiated from the rest of species molecularly analysed in the present study.

*Cratera imbiri* Araujo, Carbayo, Riutort & Álvarez-Presas, sp. nov.
urn:lsid:zoobank.org:act:CC5B22EB-9E7C-490F-A6FF-03757BA03C26

**Etymology.** The name *imbiri* refers to Vila de São Matheus do Imbiri, former name of Campos do Jordão, type locality of the species. The specific epithet is invariable.

**Type locality.** Parque Estadual Campos do Jordão, Campos do Jordão, State of São Paulo, Brazil.

**Distribution.** Only known from the type locality.

**Material examined.** Holotype MZUSP PL 2155 (Field code F5512), sexually mature: Parque Estadual Campos do Jordão, Campos do Jordão, State of São Paulo, Brazil (−22.68878, −45.48068). coll. F. Carbayo and co-workers, 15 November 2012. Horizontal sections of a body portion behind cephalic extremity on 7 slides; transverse sections of pre-pharyngeal region on 9 slides; sagittal sections of pharynx on 13 slides; sagittal sections of copulatory apparatus on 9 slides.

## Diagnosis

Species of *Cratera* 26 mm long in preserved condition; dorsal median stripe sulfur yellow, bordered on either side by a khaki grey band; body margins cream; in anterior 1/4th of the body, this pattern covered with a color gradient of coral red; eyes marginal; pharynx cylindrical, with dorsal insertion posteriorly shifted at the equivalent of 20% the length of pharynx; pharyngeal pouch very close to prostatic vesicle; paired portion of the prostatic vesicle with 1/3th of total length of this organ; epithelium of penis papilla underlain by a layer of circular muscle fibers; female atrium 3.2 times longer than male atrium; female atrium narrows gradually towards its posterior section; common glandular ovovitelline duct long.

## Description

When creeping, body 38 mm long and 2.5 mm wide. Preserved 26 mm and four mm, respectively. Body margins parallel along most of its length. Extremities of the body rounded. The dorsum slightly convex, ventral side flattened. Creeping sole, 94% of body width at the pre-pharyngeal region. Mouth at a distance from anterior extremity equal to 70% of body length; gonopore, 78%.

Dorsal color with a sulfur yellow median stripe, 14% of the body width, this bordered on either side by a khaki grey band, 34% of the body width. Body margins (9% of body width) cream (Fig. 12A). In anterior 1/4th of body, this pattern covered with a gradient color of coral red. Ventral side coral red along anterior 1/4th, and cream colored behind (Fig. 12B).

Each eye is formed by a single pigment cup 22–25 μm in diameter. No clear halos around eyes were observed. Eyes contour the anterior extremity in a single row and extend marginally until posterior extremity. Anteriormost region of the body, with three mm in length, was not available for histological examination since it was degraded for DNA extraction. Sensory pits, 18–20 μm deep, as a uniserial ventro-lateral row extending backwards along a body 3.7 mm of the body. Necks of gland cells producing erythrophil

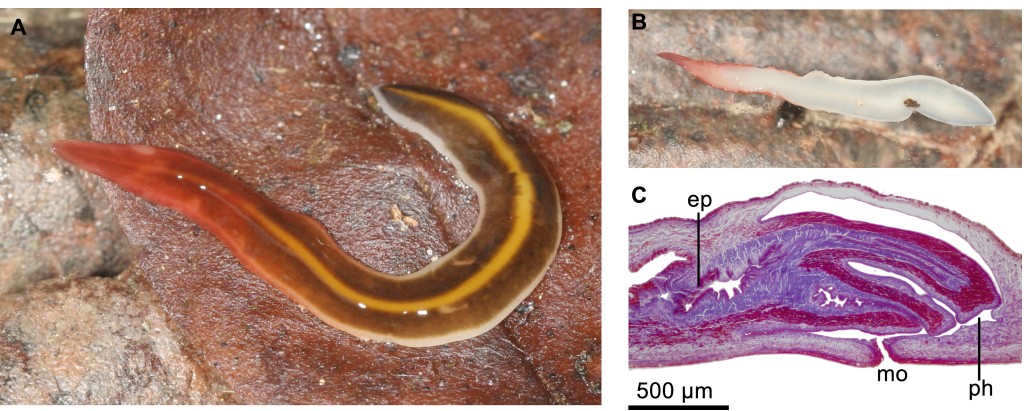

**Figure 12 *Cratera imbiri* sp. n. morphological features.** *Cratera imbiri* sp. n. Holotype (A) Dorsal view of living specimen. Scale bar not available. (B) Ventral view of living specimen. Scale bar not available. (C) Photomicrograph of a sagittal section of pharynx.

granules pierce dorsal and ventral epithelium in pre-pharyngeal region. In addition, rhabditogen cells discharge their content through dorsal epithelium. Glandular margin constituted of abundant glands producing erythrophil granules.

Cutaneous musculature formed by a subepidermal circular muscle, followed by two diagonals with decussate fibers, and a subjacent longitudinal muscle one, 35 μm thick dorsally and 30 μm thick ventrally. Thickness of cutaneous muscle coat, 11.3% of body height in the pre-pharyngeal region.

Three parenchymal muscle layers are present throughout the body: a dorsal layer of diagonal decussate fibers (10–13 μm thick), a transverse supraintestinal layer (18–24 μm), and a transverse subintestinal one (20–23 μm).

Central nervous system as a ventral nerve plate. Cerebral ganglia not discerned.

Mouth located at the end of the anterior half of pharyngeal pouch (Fig. 12C). Pharynx cylindrical, with dorsal insertion posterior to the ventral one at the equivalent of 7% of pharyngeal length. Esophagus length 20% of pharyngeal length. Outer pharyngeal epithelium underlain by a one-fiber-thick layer of longitudinal muscle fibers, followed by a layer of circular fibers (5 μm thick); inner pharyngeal epithelium underlain by a well-developed layer of circular muscle fibers (60–100 μm), followed by a thinner layer of longitudinal fibers (8–9 μm). Pharyngeal pouch 80 μm anterior to prostatic vesicle.

Testes mature, dorsal, located under the supraintestinal transverse muscle layer, partially placed between the intestinal diverticula. The testes extend from 200 μm behind the level of the ovaries to one mm anterior to the root of pharynx. Sperm ducts highly constricted at the point of communication with the respective branches of prostatic vesicle. Paired portion of this vesicle occupies ca. 1/3rd of the total length of the organ. Prostatic vesicle extrabulbar, running postero-dorsally until anterior region of penis bulb. Vesicle lined with a columnar, ciliated epithelium, which is pierced by necks of cells producing fine erythrophil granules. A coat of 50-μm-thick circular muscles surrounds the vesicle. Inside the penis papilla, vesicle communicates with the horizontal ejaculatory duct, which is sinuous proximally.

This duct is widened distally at the tip of the penis papilla at the equivalent of 2/5th of length of penis papilla. Ejaculatory duct lined with a cuboidal, ciliated epithelium, its cilia being as long as cell height, i.e., 10 μm. Ejaculatory duct surrounded by a 5-μm-thick layer of circular muscle fibers.

Cylindrical penis papilla short, horizontally orientated, with rounded tip; it is shorter than male atrium (Figs. 13A–13C). Male atrium as long as 1.2 its height, with smooth folds. A large, transverse, annular fold strongly narrows communication with female atrium (Fig. 13A, 13C). Penis papilla and male atrium clothed with a cuboidal-to-columnar, non-ciliated epithelium; the subapical portion of its cells bieng xanthophil. Papillar epithelium pierced by necks two types of cells producing granules, one erythrophil, another weakly basophil. Additionally, cells with gross necks (6 μm in diameter) and erythrophil amorphous appearance are located immediately under the epithelium. Epithelium of penis papilla and that of male atrium underlain by a 6-μm-thick layer of circular muscle fibers.

Vitellaria well developed. Ovaries ellipsoid, measuring 450 μm in diameter in antero-posterior direction (Fig. 13D), located immediately above the ventral nerve plate, at a distance from anterior tip equivalent to 21% of body length. Ovovitelline ducts arise from dorso-lateral wall of ovaries, subsequently run backwards above the ventral nerve plate. Oviducts ascend laterally to the female atrium to unite dorsally to the mid-portion of the female atrium to form the common glandular ovovitelline duct (Fig. 13A). The distal half, ascending portion of the ducts receives the openings of shell glands. The common glandular ovovitelline duct is about 0.9 mm long (47% of the length of female atrium), and communicates with the female genital canal. This canal is a projection of the postero-dorsal portion of the female atrium and is forwards and dorsally directed. Female atrium 3.2× the length of male atrium. Posterior third of female atrium with lateral folds narrowing its lumen. Female atrium lined with a non-ciliated, 20-μm-tall epithelium along anterior 3/4 of its length. Gland cells producing erythrophil granules discharge their secretion into female atrium, which seems to be underlain by two muscle layers.

## Discussion

This small species displays a color pattern that cannot be confounded with any of its congeners. Regarding the internal morphology, only *C. tui* resembles *C. imbiri* in that both species have an uncommonly long female atrium, at least 2.5 times longer than the male one. Indeed, *C. tui* and *C. imbiri* are very similar in the general aspect of the copulatory apparatus. They can be distinguished from each other a number of small anatomical details: (a) the pharyngeal pouch is 0.75 mm anterior to the prostatic vesicle (vs. practically at the same level as the prostatic vesicle in *C. imbiri*); (b) dorsal insertion of the pharynx is posteriorly shifted at the equivalent of 20% the length of pharynx (vs. 7% in *C. imbiri*); (c) paired portion of the prostatic vesicle is 1/3th of total length of this organ in *C. tui* (vs. half in *C. tui*); (d) epithelium of penis papilla is underlain by a layer of circular muscle fibers, followed by a layer of longitudinal ones in *C. tui* (vs. only a layer of circular muscle in *C. imbiri*); and (e) the female atrium narrows abruptly towards its posterior section in *C. tui* (vs. gradually in *C. imbiri*). The molecular-based phylogeny shows these two species as very

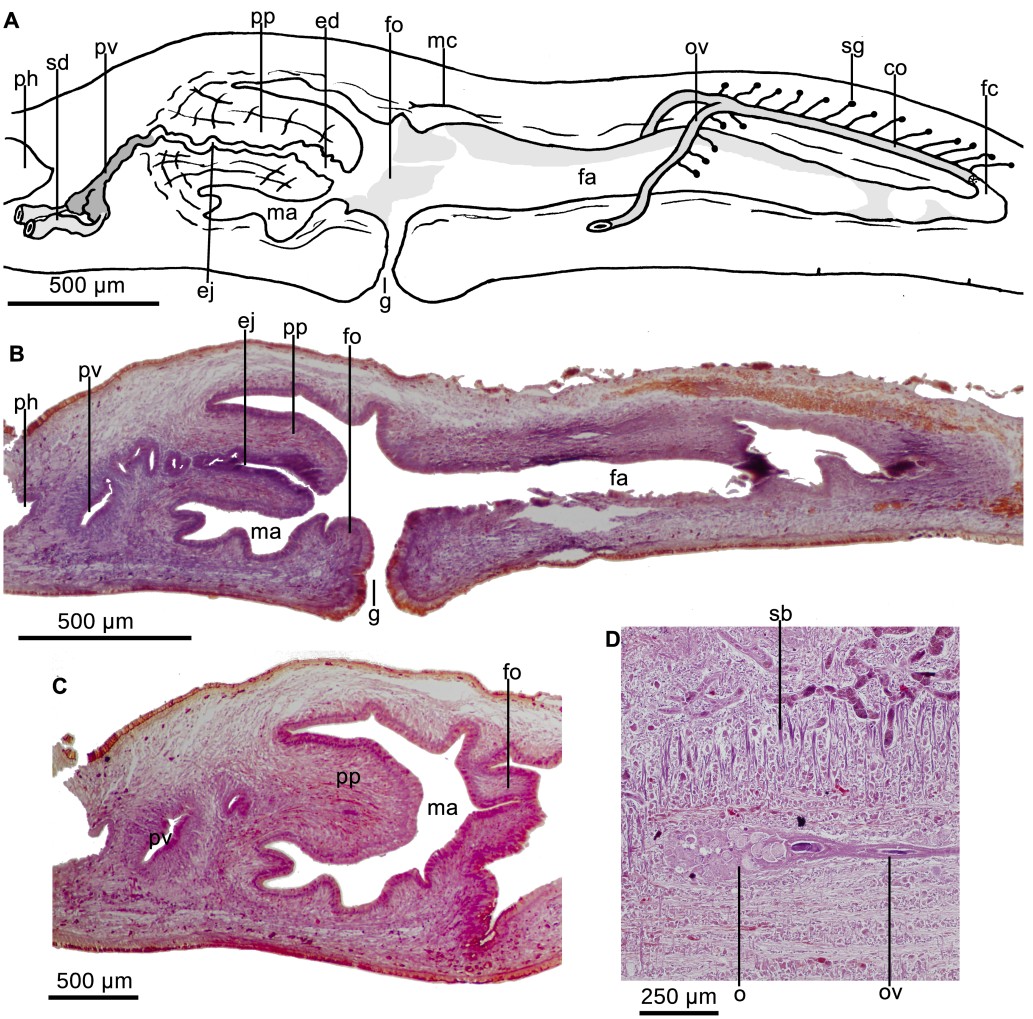

**Figure 13** *Cratera imbiri* **sp. n. morphological features.** *Cratera imbiri* sp. n. Holotype. (A) Diagrammatic representation of the copulatory apparatus from sagittal sections. (B) Photomicrograph of a sagittal section of copulatory apparatus. (C) Photomicrograph of a sagittal section of male atrium. (D) Photomicrograph of a horizontal section of ovary.

close genetically, nonetheless, the molecular delimitation identifies them as two clearly distinct species, which reinforces the small anatomical differences as being species-specific.

*Cratera paraitinga* **Araujo, Carbayo, Riutort & Álvarez-Presas, sp. nov.**
urn:lsid:zoobank.org:act:7B3F43A7-2794-42F4-B99F-B8C062F972CF

**Etymology.** The name *paraitinga* refers to São José do Paraitinga, former name of Salesópolis, type locality of the species. The specific epithet is invariable.
**Type locality.** Estação Biológica de Boraceia, Salesópolis, São Paulo State, Brazil.
**Distribution.** Type locality only.

**Material examined.** Holotype MZUSP PL 2157 (Field code F5769), sexually mature: Estação Biológica de Boraceia, Salesópolis, São Paulo State, Brazil (−23.65413, −45.88884). coll. F. Carbayo and co-workers, 20 April 2013. Transverse sections of cephalic extremity on 19 slides; horizontal sections of a portion immediately behind on 71 slides; transverse sections of pre-pharyngeal on 22 slides; sagittal sections of the pharynx on 33 slides; sagittal sections of copulatory apparatus on 60 slides. Paratype MZUSP PL 2156 (Field code F5745), incompletely mature: Ibidem. Transverse sections of cephalic extremity on 11 slides; horizontal sections of a body portion immediately behind cephalic region on 23 slides; transverse sections of pre-pharyngeal region on 7 slides; sagittal sections of pharynx and copulatory apparatus on 12 slides.

### Diagnosis

Species of *Cratera* up to 76 mm long in preserved condition; dorsal melon yellow median stripe, bordered on either side by a jet black stripe, with externally to the latter a marginal traffic white stripe; body margins jet black; eyes marginal; anterior 1/6th of the body colored with a gradient of carmine red; pharynx cylindrical to bell-shaped; pharyngeal pouch *ca.* two mm anterior to prostatic vesicle; distal dilation of ejaculatory duct relatively large; penis papilla as long as male atrium; female atrium 2.4 times longer than the male atrium; common glandular ovovitelline duct long.

### Description

The longest and only mature specimen (holotype) measured 75 mm in length, and 7 mm in width. Paratype incompletely mature, 27 mm long and four mm wide. Body slightly lanceolate, with maximum width at the level of the pharynx. Anterior to the pharynx, body narrows gradually towards the rounded, anterior tip; posterior region narrows more abruptly. Dorsum convex, ventral side flattened. Creeping sole, 92–95% of body width at pre-pharyngeal region ($n = 2$). Mouth at a distance from anterior extremity equal to 63% of body length; gonopore, 83% (holotype).

Dorsum with an melon yellow median stripe, measuring 40% of the body width, on either side bordered by a jet black stripe (14.5%), with externally to it a white band (11%); body margin (4.5%) jet black (Figs. 14A–14B). Body margins of anterior 1/5th of the body colored with a gradient of carmine red. Ventrally, body margins of anterior 1/6th orange brown, grey white behind (Fig. 14C).

Each eye is formed by a single pigmented cup, 20–25 μm in diameter. No clear halos around eyes were seen. Eyes contour the anterior extremity in a single row and extend marginally until posterior tip. Sensory pits, 15–16 μm deep, as a uniserial ventro-lateral row through a body length at least equal to 15%. Necks of two types of gland cells, producing xanthophil and erythrophil granules, respectively, pierce pre-pharyngeal region dorsally and ventrally. In addition, rhabditogen cells discharge their secretion through dorsal epidermis. Conspicuous glandular margin with abundant glands producing xanthophil granules (Fig. 15A).

Cutaneous musculature consisting of a subepidermal circular layer, followed by two diagonals with decussate fibers, and then a longitudinal one, 35–40 μm thick (paratype
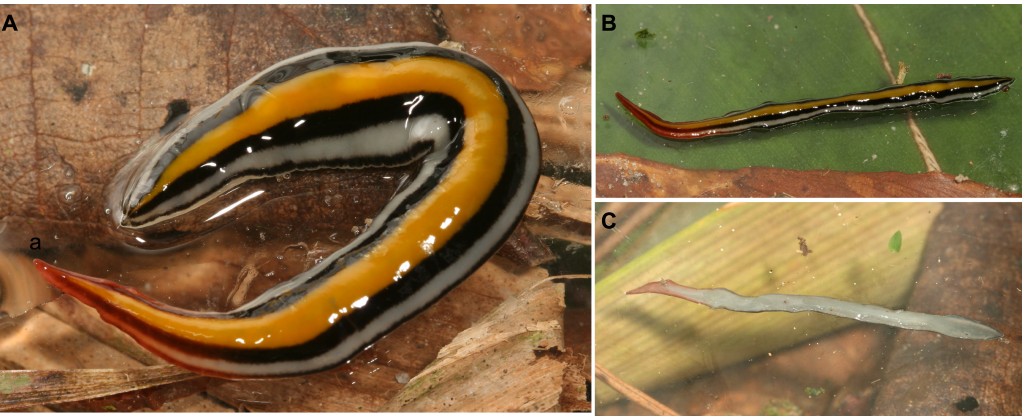

**Figure 14 *Cratera paraitinga* sp. n. morphological features.** *Cratera paraitinga* sp. n. (A) Dorsal view of living holotype. (B) Dorsal view of living paratype. (C) Ventral view of living paratype. Scales not available.

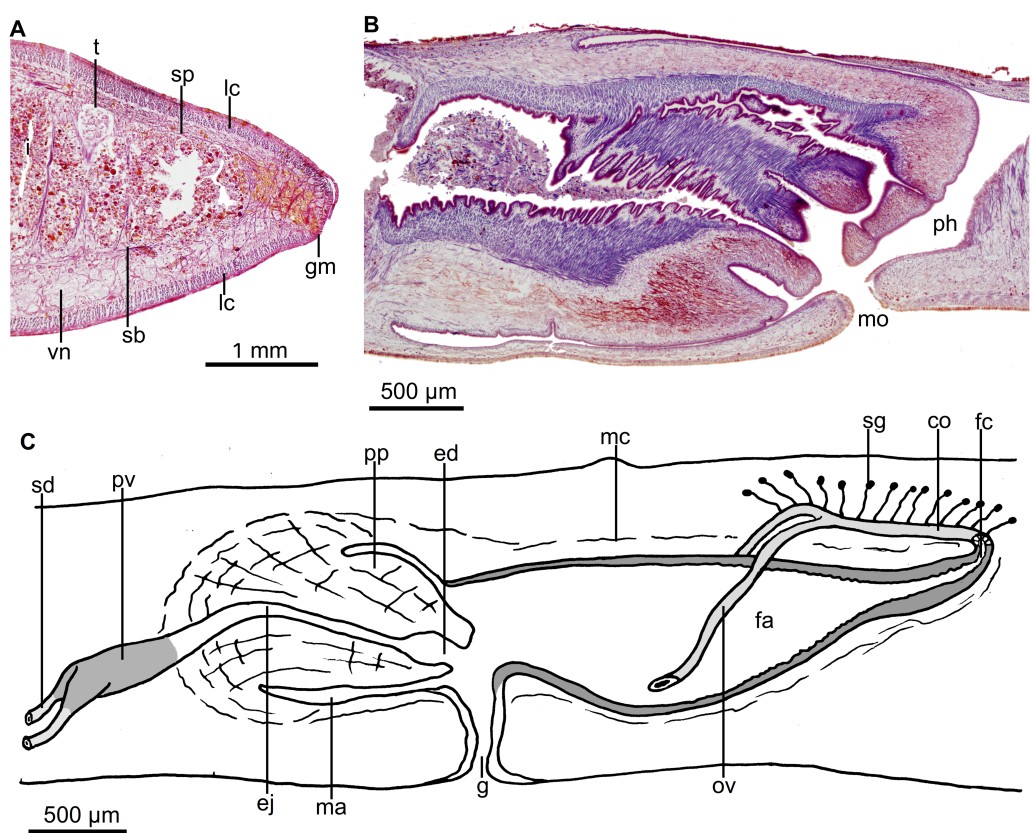

**Figure 15 *Cratera paraitinga* sp. n. morphological characters.** *Cratera paraitinga* sp. n. (A) Photomicrograph of a transverse section of pre-pharyngeal region of paratype. (B) Photomicrograph of a sagittal section of pharynx of holotype. (C) Diagrammatic representation of the copulatory apparatus of holotype from sagittal sections.

and holotype, respectively) dorsally and 40–45 μm thick ventrally (holotype and paratype, respectively). Thickness of cutaneous musculature ranging between 6.6%–10.7% ($n = 2$) to body height in the pre-pharyngeal region.

Three parenchymal muscle layers present throughout the body, all constituted by fibers relatively densely packed: a dorsal layer of diagonal decussate fibers (10 μm thick, holotype), a supraintestinal layer of transverse muscle fibers (40 μm), and a transverse subintestinal one (40 μm). Dorso-ventral fibers abundant between intestinal branches.

Central nervous system as a ventral nerve plate. Clearly evident cerebral ganglia were not observed.

Mouth located at the end of the anterior half of pharyngeal pouch (Fig. 15B). Pharynx between cylindrical and bell-shaped, with dorsal insertion posterior to the ventral one at the equivalent of 40% of pharyngeal length ($n = 2$). Esophagus length 20% of pharyngeal length ($n = 2$). Outer pharyngeal epithelium underlain by a one-fiber-thick layer (4 μm) of longitudinal muscle fibers, followed by a layer of circular fibers (6 μm); inner pharyngeal epithelium underlain by a layer (50–100 μm) of circular fibers, followed by a layer (20 μm) of longitudinal fibers (holotype). Pharyngeal pouch 2.1 mm anterior to the prostatic vesicle in holotype, and 0.65 mm in the incompletely mature paratype.

Testes dorsal, located under the supraintestinal muscle layer, partially placed between the intestinal diverticula (Fig. 15A). Testes extend from shortly behind the level of the ovaries to nearly three mm anterior to root of pharynx (holotype). Sperm ducts communicate with the respective lateral diverticula of the prostatic vesicle. Extrabulbar prostatic vesicle elongate and branched along its proximal 1/4th. Vesicle runs posteriorly and upwards until anterior region of penis bulb. Prostatic vesicle lined with a columnar, ciliated epithelium, which is pierced by numerous necks of cells producing fine erythrophil granules. A 15-μm-thick net of muscle fibers surround the vesicle. Inside the penis papilla, prostatic vesicle communicates with a straight ejaculatory duct, the latter traversing penis papilla. Ejaculatory duct widened to a conspicuous along distal half of penis papilla (Figs. 15C, 16A–16B). Ejaculatory duct surrounded by a 5-μm-thick layer of circular muscle fibers along most of its length, and by circular and longitudinal fibers on its widened portion.

Penis papilla conical and blunt, with dorsal insertion slightly posterior to the ventral one; it is as long as male atrium (Figs. 15C, 16A). Male atrium as long as 1.4× its height, slightly folded. Penis papilla and male atrium clothed with a cuboidal, non-ciliated epithelium, which is pierced by necks of cells producing erythrophil granules. Quality of stain did not allow for further details. Epithelium of penis papilla and that of male atrium underlain by a 3-μm-thick layer of circular muscle fibers, followed by a 2-μm-thick layer of longitudinal fibers.

Vitellaria well developed. Elongated, ovoid ovaries measuring 300 μm in diameter in antero-posterior direction. They are located immediately above the ventral nerve plate, at a distance from anterior tip equivalent to 25% of body length. Ovovitelline ducts arise from dorso-lateral side of ovaries and run backwards above the ventral nerve plate. Laterally to the female atrium, the oviducts curve dorso-medially and, subsequently, unite dorsally to the posterior third of the female atrium to form the common glandular ovovitelline duct (Fig. 15C). Distal ascending portion of the oviducts receives the openings of shell

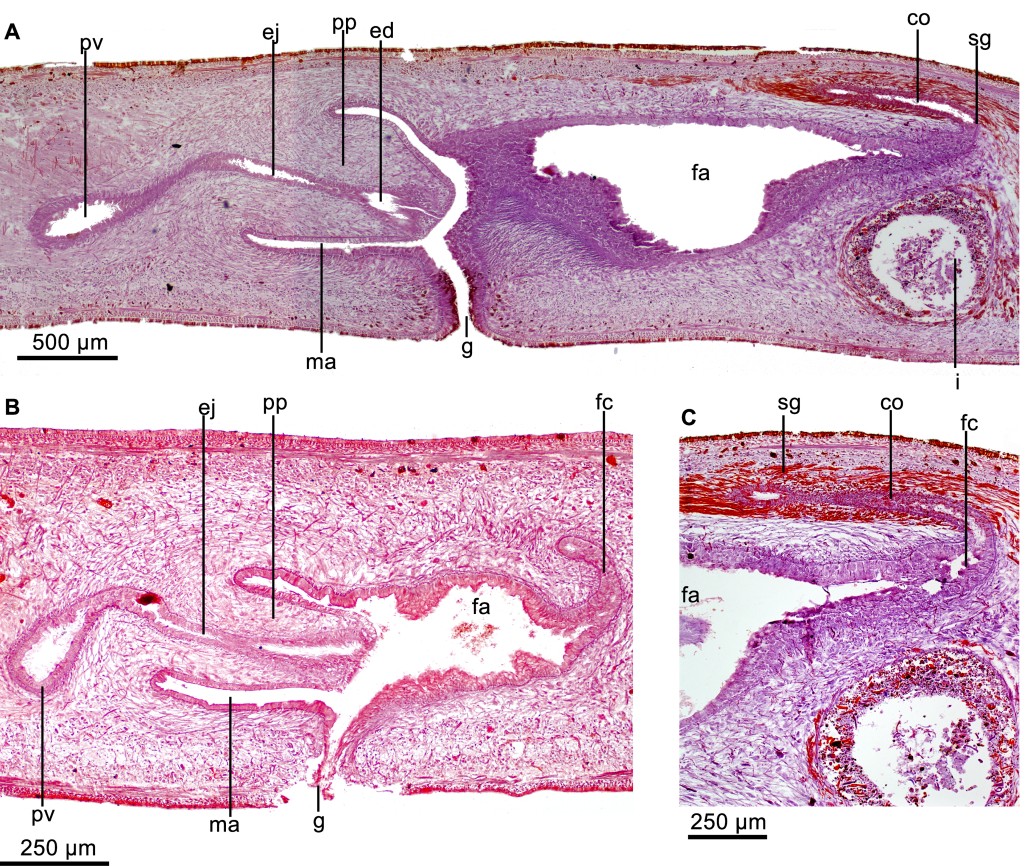

**Figure 16** *Cratera paraitinga* **sp. n. morphological features.** *Cratera paraitinga* sp. n. Photomicrographs of sagittal sections. (A) Sagittal section of copulatory apparatus of holotype. (B) Copulatory apparatus of paratype, incompletely mature. (C) Posterior region of female atrium of holotype.

glands. The long common glandular ovovitelline duct (0.9 mm, i.e., 1/3th of the length of female atrium) communicates with female genital canal, the latter being a projection of the posterior portion of female atrium that projects antero-dorsad (Figs. 15C, 16A–16C). Female atrium long and funnel-shaped, compressed laterally. Anterior portion of female atrium narrower than in its mid-region. Female atrium 2.4 times the length of male atrium, and lined with non-ciliated epithelium, 40 μm tall and with multilayered aspect at its posteriormost 1/4th; otherwise, epithelium 20 μm tall and cuboidal-to-columnar. Necks of cells producing erythrophil granules pierce female epithelium, which is underlain by a layer of circular muscle fibers (10 μm thick, holotype), followed by a layer of longitudinal fibers (10 μm thick).

## Discussion

Among all species of the genus, *C. taxiarcha* and *C. tui* resemble *C. paraitinga* in the general color pattern of the dorsum, consisting of longitudinal stripes with black, white and yellow color. However, the median stripe in *C. taxiarcha* is white (yellowish in *C. paraitinga*).

*Cratera tui*, very similar to *C. paraitinga* in the color pattern, differs from it in the width of the yellowish mid-dorsal stripe (28% of body width in *C. tui* vs. 40% in *C. paraitinga*).

Regarding the copulatory apparatus, only *C. tui* and *C. imbiri* resemble *C. paraitinga* in the relatively long female atrium as compared to the male one. However, the two species differ from *C. paraitinga* in details: (a) the dilation of the ejaculatory duct is relatively small in *C. tui* (vs. relatively large in *C. paraitinga*); (b) the penis papilla is shorter than the male atrium in *C. tui* and *C. imbiri* (vs. as long as the male atrium in *C. paraitinga*).

The molecular analyses show these three species, *C. tui*, *C. imbiri* and *C. paraitinga*, to constitute a monophyletic group, which reflects their morphological similarities but also may cast some doubt on their status as separate species. However, the discovery methods of species delimitation show the three as independent species, with the exception of the mPTP method. In the case of the validation method (BPP), the significance of the separation of *C. imbiri* and *C. paraitinga* is only highly supported by the models implying a small ancestral size, while the support is slightly lower if we consider the ancestral population as having been large. Our results could be interpreted also in the way that ancestral populations were not very large. Evidently, the current situation may be a consequence of the destruction of their habitat or a lack of sampling, since some areas have been explored very intensively and others are still pending sampling. Putting together all the evidence, molecular and morphological inferences reinforce one another and, therefore, give more weight to the small morphological differences, which therefore, should be interpreted as indicating separate species.

## GENERAL DISCUSSION

*Carbayo et al. (2013)* proposed the first phylogenetic framework of the Geoplaninae. That phylogeny was inferred from one mitochondrial region (COI) and three nuclear ones (18S, 28S rDNA and EF) of 68 putative species, eight of them representing *Cratera* lineages (plus one immature representative). At that time, only three species of *Cratera* were known (*C. crioula*, *C. pseudovaginuloides* and *C. tamoia*). Later, three of the undescribed species considered in that study were formally described (*C. cuarassu*; *C. picuia*, *C. arucuia*) (*Carbayo & Almeida, 2015*; *Lago-Barcia & Carbayo, 2018*). Two species included in the *Carbayo et al. (2013)* phylogeny that had remained morphologically unstudied, are described in the present work, namely *C. assu* and *C. tui*.

In the present study, three new species (*C. imbiri*, *C. paraitinga*, *C. boja*) are also included. The phylogenetic relationships between all these 11 species have been examined here through comparative analysis of six concatenated DNA regions (two mitochondrial fragments and four nuclear).

Without taking into consideration differences in representativeness, the topology of our phylogeny matches that of *Carbayo et al. (2013)*, except for the position of *C. tamoia* and *C. crioula*. In the phylogeny from 2013, *C. tamoia* is sister of the remaining species of an in-group including *C. crioula*, whereas in the present phylogeny, *C. crioula* + *C. assu* is the sister clade of the remaining members of the in-group, which includes *C. tamoia*. This is a relevant result, due to a difference in taxon sampling. As more species are included in

the present study, relationships are resolved that could not be observed in the phylogenies of 2013, with a smaller representation of species of *Cratera*. *Lago-Barcia & Carbayo (2018)* discussed the evolution of some morphological attributes within *Cratera* by analyzing them against the phylogeny of *Carbayo et al. (2013)*. They considered only the five species whose anatomy was known, namely *C. arucuia*, *C. crioula*, *C. cuarassu*, *C. picuia*, and *C. pseudovaginuloides*. They interpreted that the distal widening of the ejaculatory duct originated in the common ancestor of all *Cratera* members and was secondarily lost in the last common ancestor of *C. tamoia*, *C. crioula*, and *C. arucuia*. For this feature, as well as other characters (roof of the male atrium pierced by necks of numerous cyanophil glands; prostatic vesicle dorsally located; 90° rotation of the penis papilla), they concluded that "diagnostic character states of the genus can be lost or modified within recently evolved in-groups of *Cratera*, hence puzzling species classification" (*Lago-Barcia & Carbayo, 2018*).

In the light of our new phylogenetic framework, loss of the widening of the distal section of the ejaculatory duct apparently evolved independently in two lineages, thus giving rise to the condition is *C. crioula*, and that in *C. picuia* and *C. arucuia*. However, this new framework does not invalidate the above-quoted conclusion of Lago-Barcia & Carbayo. Moreover, our data corroborates their conclusion, as demonstrated in the following five selected examples. (i) The position of the prostatic vesicle, either internal to the penis bulb or external to it, appears to have independently evolved from an external to an internal position only in *C. tamoia* and *C. picuia*. An equally parsimonious interpretation would be that the internal position of the prostatic vesicle would have evolved in the common ancestor of *C. tamoia* + *C. arucuia* + *C. picuia* and that this condition would have reversed in *C. arucuia*. (ii) In similar vein, a penis papilla longer than the male atrium may have evolved in the common ancestor of *C. boja*, *C. pseudovaginuloides*, *C. crioula*, *C. assu*, *C. picuia*, *C. arucuia* and *C. tamoia*, while this condition would independently and secondarily have been lost in *C. boja* and *C. tamoia*. (iii) In *C. picuia* and *C. boja*, the dorsal surface of the male atrium is traversed by a mass of necks of cells producing cyanophil granules. This situation is best explained as two evolutionary independent acquisitions. (iv) The most parsimonious explanation for the very reduced or even absence of the common glandular ovovitelline duct in *C. picuia*, *C. assu* and *C. boja* is independent loss in each of these species, none of which shares a sister-group relationship.

Against this trend, the relatively long female atrium, in comparison with the male atrium, appears to be homologous in all members of *Cratera*. The female atrium is usually as long as the male one. However, in *C. tui*, *C. imbiri*, and *C. paraitinga*, the female atrium is >2.4 times longer than the male one. These three species constitute a monophyletic group and most probably this character state evolved in the common ancestor of these species. These three species are similar to each other, not only in this trait, but also in the traditional characteristics used in the classification of Geoplaninae. For this reason, our molecular approach in the species delimitation proved to be essential in their discovery as independent lineages.

The causes underlying the evolutionary differences between the copulatory organs in land triclads remain unclear. Absence of relevant apomorphies in other related groups, such as the freshwater planarian genus *Girardia* (Dugesiidae), also complicated assignment

of species to the genus (*Sluys, Hauser & Wirth, 1997*). In the case of land planarians, morphological differences may be related to the fact that each species belongs to a lineage that has evolved independently for a long period, as exemplified by the land planarian *Cephaloflexa bergi* (Graff, 1899) (Geoplaninae), a species that originated about 7 Mya (*Álvarez Presas et al., 2014*).

The inescapable conclusion from this argumentation is that *Cratera* land planarians apparently evolved fast-evolving features, even those that diagnose the genus, such as the dilation of the ejaculatory duct. Such rapidly evolving structures can mislead a natural classification of *Cratera* and its relatives when systematics is solely based on morphology.

An interesting aspect of land planarians is their restricted geographical distribution. Most species are known from only one or a few localities (*Carbayo & Froehlich, 2008*). In the present study only *Cratera tui* was found in an additional locality apart from the type locality. But even these two localities are only 30 km apart from each other. Although sampling artifacts may underlie such presumably restricted distributions (*Sluys, 1999*), it is also possible that they reflect actual species distributions. Our data support the latter hypothesys in the case of *Cratera* because we performed an intensive sampling effort in the four studied areas that resulted in the distributional ranges reported in this study.

We hypothesize that closely related *Cratera* species with such fast-evolving features, and very restricted areas of distribution may be the result of relatively recent speciation events linked to the postglacial history of the area. However, thorough studies, including NGS data and robust population analyses, will be necessary to test this hypothesis.

## CONCLUSIONS

Molecular-based phylogenies and species delimitation methods provide hypotheses on species recognition that are independent from the morphology-based approach. Congruence of both approaches allowed us to recognize evolutionarily independent lineages, i.e, species, and to independently evaluate small morphological differences between the individuals as a signal of separate species status. Otherwise, most likely we had ranked *C. tui*, *C. imbiri* and *C. paraitinga* under one nominal species.

Furthermore, the new molecular markers for species delimitation and phylogenetic inference developed for the first time in the present work resulted in highly resolved phylogenetic trees of terrestrial planarians. We have expanded the number of informative molecular markers by adding two new molecules (Tnuc813 and Nd4toCox1) as a result of the use of new generation molecular tools. This result should not be overlooked, since the availability of molecular markers has always been a limiting factor in the molecular systematics of these animals.

**Abbreviations used in figures**

| | |
|---|---|
| **cg** | cyanophil gland cells |
| **co** | common glandular ovovitelline duct |
| **e** | eye |
| **ed** | dilated portion of ejaculatory duct |
| **ej** | ejaculatory duct |

| ep | esophagus |
|----|-----------|
| fa | female atrium |
| fc | female genital canal |
| fo | fold |
| g | gonopore |
| gl | glands |
| i | intestine |
| lc | longitudinal cutaneous muscle fibers |
| m | muscle fiber |
| ma | male genital atrium |
| mc | common muscle coat |
| mo | mouth |
| o | ovary |
| ov | ovovitelline duct |
| ph | pharyngeal pouch |
| pb | penis bulb |
| pp | penis papilla |
| px | pharynx |
| sb | subintestinal transverse muscle fibers |
| sd | sperm duct |
| sg | shell glands |
| sp | supraintestinal transverse muscle fibers |
| t | testis |
| vi | vitellaria |
| vn | ventral nerve plate |

# ACKNOWLEDGEMENTS

We thank Celso Barbieri Junior, Cláudia Olivares, Débora Redivo, Erica Panachuk de Souza, Júlio Pedroni, Leonardo Zerbone, Marcos Santos Silva, Marília Jucá and Welton Araújo (EACH, USP) for assistance during the sampling. Thanks are due to Geison Castro da Silveira, Lucas Beltrami, and Ítalo Silva de Oliveira Souza (EACH, USP) for histological processing. Gema Blasco is thanked for the wet lab support. This paper was improved after careful revision by Ronald Sluys, Lisandro Negrete and Victor Hugo Valiati.

## Funding

Ana Paula Goulart Araujo's work was supported by a CAPES graduate fellowship. Fernando Carbayo has financial support from FAPESP (Proc. 2016/18295-5) and FAPESP (Proc. 2019/12357-7). Marta Álvarez-Presas had the support of the PDJ-2014 grant from Agència de Gestió d'Ajuts Universitaris i de Recerca (AGAUR). This work was supported by the Ministerio de Economía y Competitividad of Spain (projects CGL2015–63527-1P and CGL2011–23466). The funders had no role in study design, data collection and analysis, decision to publish, or preparation of the manuscript.

### Grant Disclosures

The following grant information was disclosed by the authors:
CAPES.
FAPESP: 2016/18295-5, 2019/12357-7.
Agència de Gestió d'Ajuts Universitaris i de Recerca (AGAUR): PDJ-2014.
Ministerio de Economía y Competitividad of Spain: CGL2015–63527-1P, CGL2011–23466.

### Competing Interests

Marta Riutort is an Academic Editor for PeerJ.

### Author Contributions

- Ana Paula Goulart Araujo performed the experiments, analyzed the data, prepared figures and/or tables, authored or reviewed drafts of the paper, and approved the final draft.
- Fernando Carbayo and Marta Álvarez-Presas conceived and designed the experiments, performed the experiments, analyzed the data, prepared figures and/or tables, authored or reviewed drafts of the paper, and approved the final draft.
- Marta Riutort conceived and designed the experiments, prepared figures and/or tables, authored or reviewed drafts of the paper, and approved the final draft.

### Field Study Permissions

The following information was supplied relating to field study approvals (i.e., approving body and any reference numbers):

Field experiments were approved by COTEC - Instituto Florestal do Estado de São Paulo (Proc. SMA 12.640/2011), Museu de Zoologia (EBBAut.020/2013) and Instituto Chico Mendes de Conservação da Biodiversidade (Proc. 32779-1; 11748-4).

### DNA Deposition

The following information was supplied regarding the deposition of DNA sequences:

The Cox1 sequences are available at GenBank: MT437763 to MT437784.
The 18S sequences are available at GenBank: MT441685 to MT441699.
The 28S sequences are available at GenBank: MT441708 to MT441722.
The EF sequences are available at GenBank: MT468577 to MT468591.
The Tnuc813 sequences are available at GenBank: MT468592 to MT468620.
The Nad4Cox1 sequences are available at GenBank: MT468621 to MT468635.

### Data Availability

Data is available at Figshare: Álvarez-Presas, Marta; Araujo, Ana Paula G.; Carbayo, Fernando; Riutort, Marta (2020): Sequences from Cratera new species. figshare. Dataset. https://doi.org/10.6084/m9.figshare.12231080.v1.

### New Species Registration

The following information was supplied regarding the registration of a newly described species:

Publication LSID: urn:lsid:zoobank.org:pub:F6B30CB7-6114-434F-9B2A-A2F4CE625A20

Cratera paraitinga Araujo, Carbayo, Riutort & Álvarez-Presas, sp. nov. LSID: urn:lsid:zoobank.org:act:7B3F43A7-2794-42F4-B99F-B8C062F972CF

Cratera boja Araujo, Carbayo, Riutort & Álvarez-Presas, sp. nov. LSID: urn:lsid:zoobank.org:act:46D76DD7-B129-461D-8803-3E918AA4601C

Cratera assu Araujo, Carbayo, Riutort & Álvarez-Presas, sp. nov. LSID: urn:lsid:zoobank.org:act:DE3D812D-C387-40BD-9273-7BC1FE59D09C

Cratera tui Araujo, Carbayo, Riutort & Álvarez-Presas, sp. nov. LSID: urn:lsid:zoobank.org:act:FB96BE96-86AD-41FB-961F-C0337C56A596

Cratera imbiri Araujo, Carbayo, Riutort & Álvarez-Presas, sp. nov. LSID: urn:lsid:zoobank.org:act:CC5B22EB-9E7C-490F-A6FF-03757BA03C26

## Supplemental Information

Supplemental information for this article can be found online at http://dx.doi.org/10.7717/peerj.9726#supplemental-information.

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
