# Peer review of "Five new pseudocryptic land planarian species of Cratera (Platyhelminthes: Tricladida) unveiled through integrative taxonomy"

_PeerJ, doi:10.7717/peerj.9726_

## Round 0.1 · original submission · Major Revisions

This is certainly an important paper on systematics of land flatworms. The strong points are the use of new molecular markers, with a total of 5, and different methods of molecular species delimitation.

The paper was reviewed by 3 reviewers; Reviewer 1 has provided a detailed account; Reviewers 2 and 3 provided shorter accounts; all three reviewers sent an annotated version of the original manuscript, sometimes with many comments.

With all these corrections and modifications on a 85-page manuscript, I finally considered that the manuscript needs major modification.

These are my own remarks, to be considered as if I was Reviewer 4.

Abstract:
“Atlantic forest” – it would not hurt to add “Brazil” here.
“five nuclear and mitochondrial genes” – you can probably enumerate them.
“young species” – perhaps using the term “recent speciation events” would be more appropriate.

These are my Editorial comments concerning the strong criticisms by Reviewer 1 on the Latin names.

Reviewer 1 is correct in that the names of the new species could be built on more classical grounds, i.e. Latin names with proper endings and I agree with this opinion. However, Recommendation 25C is a Recommendation, not a Rule. In contrast, Article 11.3 reads: “Derivation. Providing it meets the requirements of this Chapter, a name may be a word in or derived from Latin, Greek or any other language (even one with no alphabet), or be formed from such a word. It may be an arbitrary combination of letters providing this is formed to be used as a word”. There are other names of geoplanid species which follow the rules used by the present author, a famous one being Platydemus manokwari described by de Beauchamp – not “manokwariensis” as it should have been, and that was in 1962.
If the authors prefer to keep their names, they should add after each etymology a clear mention that the species name is invariable.

I am also deeply concerned by the comments about describing a new species from only the holotype when many specimens are available.

·

Basic reporting

Generally fine, but see the General Comments for the author.

Experimental design

Fine. See the General Comments for the author

Validity of the findings

Fine. See the General Comments for the author.

Additional comments

Reviewer: Ronald Sluys

Manuscript (#48478) for PeerJ

Five new pseudocryptic land planarian species of Cratera (Platyhelminthes: Tricladida) unveiled through integrative taxonomy

by Araujo, Carbayo, Riutort & Álvarez-Presas

General comments:
This is an exemplary manuscript on an integrative taxonomic study, combining morphological and molecular information. All figures are good and necessary. Although the writing is not bad at all, there still remains much to be desired. Although I think that this is not actually the task of a reviewer, I have inserted very many suggestions for linguistic improvements, as well as other comments (a total of 632). I hope that these suggestions are clear and self-explanatory. There are a few other, major, issues that I would like to address below.

I was disappointed to notice that the species descriptions mostly focus on describing the holotype, even in cases where there are ample paratypes. This is absolutely incorrect, as one has to describe the species on the basis of all specimens available. This means that one has to integrate all information into a single account of the species and, thus, one provides ranges and averages of measurements. In contrast, the present species accounts give absolute measurements. It does not only concern measurements, of course, but also description of the shape of structures, etc. As Mayr (1969) already wrote: ''Description of a new species is based on the entire material available to the zoologist, including the type specimen[s]. It is NOT the function of the type to serve as the exclusive or primary base of the description''. In the case that there is only one specimen available (the holotype) one cannot give ranges, of course. Nevertheless, in the diagnosis one should avoid such absolute statements as, for example, "body length 36 mm" because one knows that other specimens will differ in details. In such cases the statement is easily made more appropriate by writing "body length about 36 mm". For other characters insertion of words such as "more or less" serve the same function.

I distinctly remember that I discussed this issue several years ago with one of the authors of the manuscript. Then he appeared to be fully convinced that indeed this focus on type specimen(s) was completely wrong and he fully agreed with Mayr’s (1969) statement. The present manuscript tells differently and, therefore, should be corrected, as in this respect it fails in its integrative objective.

Second, I have some doubts about the appropriateness and correctness of the new specific epithets coined by the authors. The International Code of Zoological Nomenclature (1999) specifies in Recommendation 25C: "Authors should exercise reasonable care and consideration in forming new names to ensure that they are chosen with their subsequent users in mind and that, as far as possible, they are appropriate, compact, euphonious, memorable, and do not cause offence."

In this respect, I feel that the epithets piguaiassu, piguaiatui, piguaiaboja, and, to lesser extent, paraitinga, fail to be compact, euphonious or memorable. Certainly, the first three mentioned are unpronounceable and should be replaced. Furthermore, the first three names mentioned, as well as the epithet imbiri, do not agree in gender with the genus name Cratera, an absolute requirement of the Zoological Code. The unpronounceable names are based on word compositions and, therefore, should agree in gender with the genus (by accident one of these epithets agrees). I do strongly suggest that the authors devise new word compositions based on euphonious Latin words that agree in gender with Cratera.

The specific epithets imbiri and paraitinga concern geographical names. Derivation and declension of geographical names are complex. In general, a specific epithet derived from a geographical name should preferably be (a) an adjective (ending in e.g., -ensis), (b) a noun in the genitive case. Correct geographic epithets are, for example, orientalis, arizonicus, atlanticus, sanctaehelenae.

Next, it is strange that the manuscript uses two different codes for the specimens, i.e., personal or field codes (I guess that these are the F codes) as well as proper museum catalogue numbers. In a published paper only official registration codes of the slides/preparations should be mentioned. Evidently, the field codes will be included in the catalogue of the museum but such field codes are of no concern or value to the reader of the paper. Presumably, the field codes are also written on the labels of the slides, as well as the proper catalogue numbers. But again, this is of no importance for the reader. Personally, I do always include field codes and proper museum catalogue numbers up to the penultimate version of a manuscript (in order to ensure that no mistakes will be made in reference to the slides) but then I remove all field codes from the final version of the manuscript (but the field codes are included in the collections database). However, in case something else is implied by the F-codes, this should be made clear in the text of the manuscript.

In the legends to the figures, the various frames of a single figure are labeled with capital letters (A, B, C, etc.), whereas in the frames themselves the indications are lower case (a, b, c, etc.); this is inconsistent; replace with capitals.

Detailed comments:
For detailed comments I refer to the annotated pdf version of the manuscript (hereby attached).

·

Basic reporting

No comments.

Experimental design

No comments.

Validity of the findings

No comments.

Additional comments

The authors provide an interesting article about the diversity of the genus Cratera (Platyhelminthes: Geoplanidae), with the description of five new species.
The descriptions of these new entities are supported by morphological features and also validated by molecular approaches. The authors selected two mitochondrial and four nuclear markers for molecular analyses, two of them (Nd4toCox1, Tnuc813) are tested for the first time in phylogenetic inferences of land planarians. The development of these new markers is very good news, mainly for those who study the phylogeny of triclads. The figures and table (plus supporting information) are informative and of good quality.
The manuscript by Araujo et al. adds more information to the genus and, according to the new phylogenetic framework, establish the relativity of some features used in the diagnosis of Cratera.
In summary, I consider that the paper can be accepted for publication in PeerJ with some minor changes.
The authors will find my comments and suggestions in the manuscript.

·

Basic reporting

No comment

Experimental design

No comment

Validity of the findings

No comment

Additional comments

This is a good manuscript, providing ample argumentation and documentation of five new species of land flatworms from South America. The authors use an integrative approach with very robust molecular and morphological data. The text is clear in its objectives, the methodology is appropriate, tables and figures are well presented, and the discussion contemplates the purposes of the work. The authors found some minor corrections and suggestions throughout the manuscript. Considering this fact, I have no further considerations and therefore my recommendation is that the article can be published with the small considerations highlighted in the text.

---

## Round 0.2 · accepted · Accept

Please receive my congratulations on this excellent paper, and also [special note for the last author] congratulations on a new personal achievement!

·

Basic reporting

Good

Experimental design

Good

Validity of the findings

Good

Additional comments

The authors have made a great effort in improving their manuscript by incorporating the numerous suggestions made by the reviewer. For this I wish to express my compliments.